# *Toxoplasma* bradyzoites exhibit physiological plasticity of calcium and energy stores controlling motility and egress

Yong Fu[1], Kevin M Brown[1†], Nathaniel G Jones[1‡], Silvia NJ Moreno[2], L David Sibley[1]*

[1]Department of Molecular Microbiology, Washington University in St. Louis, School of Medicine, St Louis, United States; [2]Center for Tropical and Emerging Global Diseases and Department of Cellular Biology, University of Georgia, Athens, United States

*For correspondence:
sibley@wustl.edu

Present address: [†]Department of Microbiology and Immunology, University of Oklahoma Health Sciences Center, College of Medicine, Oklahoma City, United States; [‡]York Biomedical Research Institute, Department of Biology, University of York, Heslington, United Kingdom

Competing interest: The authors declare that no competing interests exist.

**Abstract** *Toxoplasma gondii* has evolved different developmental stages for disseminating during acute infection (i.e., tachyzoites) and establishing chronic infection (i.e., bradyzoites). Calcium ion ($Ca^{2+}$) signaling tightly regulates the lytic cycle of tachyzoites by controlling microneme secretion and motility to drive egress and cell invasion. However, the roles of $Ca^{2+}$ signaling pathways in bradyzoites remain largely unexplored. Here, we show that $Ca^{2+}$ responses are highly restricted in bradyzoites and that they fail to egress in response to agonists. Development of dual-reporter parasites revealed dampened $Ca^{2+}$ responses and minimal microneme secretion by bradyzoites induced in vitro or harvested from infected mice and tested ex vivo. Ratiometric $Ca^{2+}$ imaging demonstrated lower $Ca^{2+}$ basal levels, reduced magnitude, and slower $Ca^{2+}$ kinetics in bradyzoites compared with tachyzoites stimulated with agonists. Diminished responses in bradyzoites were associated with downregulation of $Ca^{2+}$-ATPases involved in intracellular $Ca^{2+}$ storage in the endoplasmic reticulum (ER) and acidocalcisomes. Once liberated from cysts by trypsin digestion, bradyzoites incubated in glucose plus $Ca^{2+}$ rapidly restored their intracellular $Ca^{2+}$ and ATP stores, leading to enhanced gliding. Collectively, our findings indicate that intracellular bradyzoites exhibit dampened $Ca^{2+}$ signaling and lower energy levels that restrict egress, and yet upon release they rapidly respond to changes in the environment to regain motility.

## Editor's evaluation

This study shows that calcium signaling is strongly suppressed in the intracellular cyst-forming bradyzoite stages of *Toxoplasma gondii*. However, calcium signaling in these stages is rapidly restored following parasite egress and exposure to extracellular calcium and energy sources. The ability of *Toxoplasma* bradyzoites to rapidly switch between quiescence and a metabolically active state is likely to be essential for maintaining long-lived chronic infections as well as successful transmission.

## Introduction

*Toxoplasma gondii* is an obligate intracellular parasite, capable of infecting nearly all warm-blooded animals and frequently causing human infections (*Dubey, 2010*). The ingestion of tissue cysts in undercooked meat or shed oocysts by infected cats are the major transmission routes of *T. gondii* (*Jones and Dubey, 2012*; *Jones and Dubey, 2010*). Following oral ingestion of bradyzoites within tissue

cysts, or sporozoites within oocysts, the parasite migrates across the intestinal epithelial barrier and disseminates throughout the body as the actively proliferating tachyzoite form that infects many cell types but primarily traffics in monocytes (*Drewry and Sibley, 2019*). In response to immune pressure, the parasite differentiates to asynchronously growing bradyzoites within cysts that can persist as chronic infections in muscle and brain tissues (*Watts et al., 2015*; *Jeffers et al., 2018*; *Mayoral et al., 2020*).

Tachyzoites are adapted for rapid proliferation and dissemination due to an active lytic cycle that is controlled at numerous stages by intracellular calcium ion ($Ca^{2+}$) signaling (*Lourido and Moreno, 2015*). Artificially elevating intracellular $Ca^{2+}$ using ionophores triggers secretion of microneme proteins, which are needed for substrate and cell attachment, and hence critical for both gliding motility and cell invasion (*Carruthers and Sibley, 1999b*; *Carruthers et al., 1999c*; *Wetzel et al., 2004*). Increase of cytosolic $Ca^{2+}$ released from internal stores is sufficient to trigger microneme secretion (*Lovett et al., 2002*), and necessary for host cell invasion (*Lovett et al., 2002*; *Vieira and Moreno, 2000*), although these processes are also enhanced by the presence of extracellular $Ca^{2+}$ (*Pace et al., 2014*). Increases in intracellular $Ca^{2+}$ also precede egress and drive secretion of perforin-like protein 1 (PLP1) from microneme to facilitate rupture of parasitophorous vacuole membrane (PVM) followed by egress (*Kafsack et al., 2009*). Calcium signaling is initiated by cyclic guanosine monophosphate (cGMP)-generating guanylate cyclase (GC) (*Brown and Sibley, 2018*; *Bisio et al., 2019*; *Yang et al., 2019*) that activates parasite plasma membrane-associated protein kinase G (PKG) (*Brown et al., 2017*), stimulating the production of inositol triphosphate ($IP_3$) by phosphoinositide-phospholipase C (PI-PLC) and leading to subsequent release of intracellular $Ca^{2+}$ (*Lovett et al., 2002*; *Fang et al., 2006*; *Bullen et al., 2016*). Recent studies in *Plasmodium* also implicate PKG in directly controlling $Ca^{2+}$ through interaction with a multimembrane spanning protein that may function as a channel that mediates $Ca^{2+}$ release (*Balestra et al., 2021*). In turn, $Ca^{2+}$ activates downstream $Ca^{2+}$-responsive proteins including $Ca^{2+}$-dependent protein kinases such as CDPK1 (*Lourido and Moreno, 2015*) and CDPK3 (*Lourido et al., 2012*; *McCoy et al., 2012*), C2 domain-containing $Ca^{2+}$ binding proteins (*Tagoe et al., 2021*), and $Ca^{2+}$ binding orthologues of calmodulin (*Long et al., 2017*), which are required for invasion and egress by tachyzoites. Following invasion, protein kinase A catalytic domain 1 (PKAc1) dampens cytosolic $Ca^{2+}$ by suppressing cGMP signaling and reducing $Ca^{2+}$ uptake (*Jia et al., 2017*; *Uboldi et al., 2018*). Collectively, the lytic life cycle of tachyzoites is orchestrated spatially and temporally by controlling levels of intracellular $Ca^{2+}$ and cyclic nucleotides (*Brown et al., 2019*).

*Toxoplasma* has evolved elaborate mechanism to control intracellular $Ca^{2+}$ levels through the concerted action of $Ca^{2+}$ channels, transporters, and $Ca^{2+}$ pumps expressed at the PM and intracellular stores (*Lourido and Moreno, 2015*; *Hortua Triana et al., 2018*). Orthologues to voltage-dependent $Ca^{2+}$ channels, transient receptor potential (TRP) channels, and plasma membrane type $Ca^{2+}$-ATPases (PMCAs) are predicted to be present in *T. gondii* and likely involved in regulating cytosolic $Ca^{2+}$ influx and efflux (*Nagamune and Sibley, 2006*; *Prole et al., 2011*). The endoplasmic reticulum (ER) is an important storage site from which $Ca^{2+}$ is released to stimulate motility and egress of *Toxoplasma* (*Lourido and Moreno, 2015*). SERCA-type $Ca^{2+}$ ATPase is the known mechanism for $Ca^{2+}$ uptake by the ER, and its activity, which is inhibited by thapsigargin (*Thastrup et al., 1990*), leads to accumulation of $Ca^{2+}$ in the ER, while $Ca^{2+}$ released from the ER activates microneme secretion and motility (*Nagamune et al., 2007*; *Moreno and Zhong, 1996*). TgA1 a plasma membrane type $Ca^{2+}$ ATPase, transport $Ca^{2+}$ to the acidocalcisome (*Luo et al., 2004*), which likely provides a $Ca^{2+}$ sink albeit one that may not be as readily mobilizable as the ER. In addition to internal $Ca^{2+}$ stores, intracellular and extracellular *T. gondii* tachyzoites are capable of taking up $Ca^{2+}$ from host cells and the extracellular environment, respectively, to enhance $Ca^{2+}$ signaling pathways (*Pace et al., 2014*; *Vella et al., 2021*). A variety of fluorescent $Ca^{2+}$ indicators that have been developed to directly image $Ca^{2+}$ signals in live cells include $Ca^{2+}$-responsive dyes and genetically encoded indicators (*Vella et al., 2020*). Indicators like Fluo-4/AM, and related derivatives, have been previously used to monitor $Ca^{2+}$ levels in extracellular parasites (*Nagamune et al., 2007*; *Lovett and Sibley, 2003*). Genetically encoded $Ca^{2+}$ indicators such as GCaMP5, GCaMP6f, and GCaMP7 have also been used to visualize dynamic $Ca^{2+}$ signals of both intracellular and extracellular tachyzoites with high resolution and sensitivity (*Vella et al., 2021*; *Sidik et al., 2016*; *Borges-Pereira et al., 2015*; *Brown et al., 2016*).

In contrast to tachyzoites, little is known about the roles of $Ca^{2+}$ signaling in control of microneme secretion, gliding motility, and egress by bradyzoites. Although bradyzoites divide asynchronously,

they undergo growth, expansion, and sequential rounds of tissue cyst formation and rupture that maintain chronic infection in vivo (*Watts et al., 2015*). Histological studies in animal models support a model of periodic cyst rupture (*Ferguson et al., 1989*), releasing bradyzoites that reinvade new host cells to generate secondary daughter cysts (*Frenkel and Escajadillo, 1987*), or transition back to actively replicating tachyzoites (*Hofflin et al., 1987*). Development of bradyzoites has been studied in vitro using systems that induce development due to stress induced by alkaline pH (*Soete et al., 1993*) or in cell lines where development occurs spontaneously (*Swierzy and Lüder, 2015*; *Halonen et al., 1996*). Although numerous studies have focused on the determinants that control stage conversion between tachyzoites and bradyzoites (*Jeffers et al., 2018*; *White et al., 2014*), few studies focus on the signaling pathways that control the bradyzoite lytic cycle.

In the present study, we combined stage-specific bradyzoite fluorescent reporters with $Ca^{2+}$ imaging probes to explore $Ca^{2+}$ signaling, microneme secretion, motility, and egress by bradyzoites. Our findings indicate that bradyzoites exhibit dampened $Ca^{2+}$ levels, reduced microneme secretion, and minimal egress in response to $Ca^{2+}$ agonists. Ratiometric $Ca^{2+}$ imaging demonstrated lower $Ca^{2+}$ basal levels and significantly lower stored $Ca^{2+}$ in ER and acidocalcisome in bradyzoites, associated with reduced expression of $Ca^{2+}$ ATPases responsible for maintaining intracellular stores. Incubation of extracellular bradyzoites in $Ca^{2+}$ plus glucose leads to rapid recovery of both intracellular $Ca^{2+}$ and ATP levels and restored motility. Collectively, our findings support a dampened lytic cycle in bradyzoites, arising from diminished $Ca^{2+}$ signaling and lowered energy stores, and that upon release they exhibit rapid metabolic responsiveness to environmental conditions.

## Results
### $Ca^{2+}$ signaling triggers inefficient egress by bradyzoites

To define egress by bradyzoites, we induced the differentiation of tachyzoites to bradyzoites by culture in human foreskin fibroblast (HFF) cells at alkaline pH (8.2) for 7 days. We treated both tachyzoite cultures and in vitro-differentiated cysts with $Ca^{2+}$ ionophore A23187 to trigger egress from parasitophorous vacuoles (PVs) or bradyzoite cysts, as detected by indirect immunofluorescence assay (IFA) or time-lapse video microscopy. We observed that A23187 induced complete egress of tachyzoites from disrupted PVs while only a few bradyzoites were released from cysts that remained largely intact (*Figure 1A*). This result was also confirmed by time-lapse video microscopy using the ME49 BAG1-mCherry strain either grown as tachyzoites (*Figure 1—video 1*) or bradyzoites (*Figure 1—video 2*). We quantified the percentage of tachyzoites or bradyzoites that were released during egress in response to A23187 or the agonist zaprinast, which is a cGMP-specific phosphodiesterase (PDE) inhibitor that activates PKG-mediated $Ca^{2+}$ signaling, leading to egress. In contrast to tachyzoites, we found significantly lower egress rate of bradyzoites in response to A23817 or zaprinast (*Figure 1B*). To examine the behavior of released parasites, we determined the maximum egress distance that parasites moved away from the original vacuole or cyst following egress. Tachyzoites migrated much further than bradyzoites after induced egress (*Figure 1C*). Bradyzoites also moved more slowly than tachyzoites (*Figure 1D*), as shown by quantification of their trajectories from time-lapse video microscopy images. Taken together, these findings indicate that egress by bradyzoites in response to $Ca^{2+}$ ionophore or zaprinast is incomplete and restricted.

### Calcium-mediated microneme secretion is dampened by bradyzoite development

Egress by parasites requires $Ca^{2+}$-stimulated microneme secretion. To examine the reason for inefficient egress by bradyzoites, we monitored microneme secretion by quantitative secretion analysis of MIC2 fused with *Gaussia* luciferase (Gluc). The *MIC2-Gluc* reporter was randomly integrated into the genome of the BAG1-mCherry strain (*Figure 2A*). IFA revealed that MIC2-Gluc was expressed and localized to the apical pole in tachyzoites and bradyzoites induced for 7 days at pH 8.2 in vitro (*Figure 2B*). We also confirmed the expression of MIC2-GLuc and MIC2-associated protein M2AP, which forms a protein complex with MIC2 (*Huynh et al., 2003*), by western blotting. Although both proteins were readily detectable, they were expressed at lower levels in bradyzoites compared with tachyzoites (*Figure 2C*). Furthermore, IFA demonstrated that MIC2 and M2AP were co-localized to the apical region in bradyzoites, consistent with being located in micronemes (*Figure 2—figure*

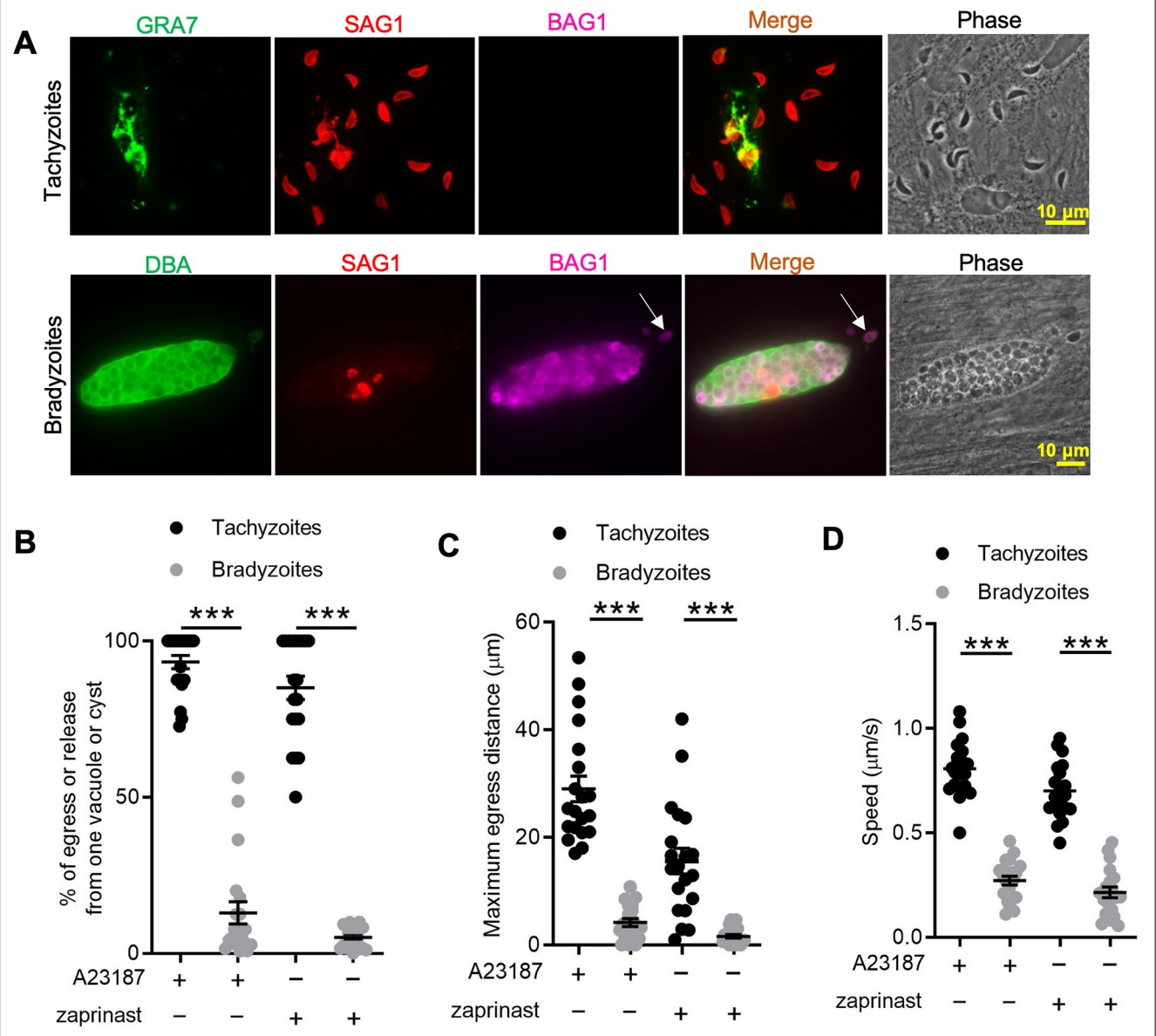

**Figure 1.** In vitro-induced bradyzoites show limited egress in response to $Ca^{2+}$ agonists. (**A**) Egress of tachyzoites and bradyzoites in response to A23187 (2 µM) for 15 min. Anti-GRA7, anti-SAG1, and anti-BAG1 antibodies followed by secondary antibodies to Alexa conjugated fluorochromes were used to detect the parasitophorous vacuole (PV) membrane, tachyzoites, and bradyzoites, respectively. *Dolichos biflorus* agglutinin (DBA) conjugated to FITC was used to stain the cyst wall. Arrow indicates released bradyzoites. Scale bar = 10 µm. (**B**) Quantitative analysis of egress in response to A23187 (2 µM) or zaprinast (500 µM) in extracellular buffer (EC) with $Ca^{2+}$ for 15 min. Each data point represents the percentage of egressed or released parasites from one PV or cyst (n = 20). Means ± SD of two independent experiments with 20 replicates. Two-tailed Mann–Whitney test, ***p<0.001. (**C**) Quantitative analysis of maximum distance egressed or released parasites moved away from the vacuole/cyst in response to A23187 (2 µM) or zaprinast (500 µM) in EC buffer with $Ca^{2+}$ for 15 min. Each data point represents distance traveled of one egressed tachyzoite or released bradyzoite from the original PV or cyst (n = 20). Means ± SD of two independent experiments with 20 replicates. Two-tailed Mann–Whitney test, ***p<0.001. (**D**) Quantitative analysis of speed (µm/s) of egressed or released parasites in response to A23187 (2 µM) or zaprinast (500 µM) in EC buffer with $Ca^{2+}$ for 15 min by time-lapse microscopy. Mean speed was determined by time-lapse recording during the first 1 min after egress or release. Each data point represents migration speed of a single egressed tachyzoites or released bradyzoites from original PV or cyst (n = 20). Means ± SD of two independent experiments with 20 replicates. Two-tailed unpaired Student's *t*-test, ***p<0.001.

The online version of this article includes the following video and figure supplement(s) for figure 1:

**Source data 1.** Percentage of egress or release from one vacuole or cyst (related to *Figure 1B*).

**Source data 2.** Maximum egress distance of egressed or released parasites (related to *Figure 1C*).

**Source data 3.** Speed of egressed or released parasites (related to *Figure 1D*).

*Figure 1 continued on next page*

*Figure 1 continued*

**Figure 1—video 1.** Egress by ME49 BAG1-mCherry tachyzoites in response to A23187.
https://elifesciences.org/articles/73011/figures#fig1video1

**Figure 1—video 2.** Egress by ME49 BAG1-mCherry bradyzoites in response to A23187.
https://elifesciences.org/articles/73011/figures#fig1video2

---

*supplement 1A*). Next, to enrich highly purified MIC2-GLuc reporter parasites, BAG1-mCherry MIC2-GLuc strain tachyzoites, and bradyzoites liberated from cysts produced by cultivation for 7 days at pH 8.2 in vitro, were sorted by fluorescence-activated cell sorting (FACS) (*Figure 2—figure supplement 1B*). FACS-sorted tachyzoites and bradyzoites were treated with zaprinast or ionomycin, a $Ca^{2+}$ ionophore that induces release of $Ca^{2+}$ from the ER (*Beeler et al., 1979*). Bradyzoites secreted much less MIC2-Gluc protein compared to tachyzoites in response to $Ca^{2+}$ agonists, zaprinast and ionomycin, as shown by *Gaussia* luciferase assays performed on excretory-secretory antigen (ESA) fractions collected following stimulation (*Figure 2D*). To further investigate the process of microneme secretion by bradyzoites, we randomly integrated an mCherry secretion reporter, based on the signal peptide sequence of ferredoxin-NADP(+)-reductase (FNR-mCherry), into the genome of BAG1-EGFP parasites (*Figure 2E*). The FNR-mCherry reporter is an improved version of DsRed reporter that is secreted constitutively and released from the PV surrounding tachyzoites following the discharge of PLP1 in response to $Ca^{2+}$ agonists (*Kafsack et al., 2009*). Then, we monitored the permeabilization of PV membrane surrounding either tachyzoite vacuoles or in vitro-differentiated bradyzoites after stimulation with A23187 based on the diffusion of FNR-mCherry using time-lapse fluorescence video microscopy. Consistent with previous reports (*Lourido et al., 2010*), we observed that A23187 stimulated fast leakage of FNR-mCherry from the PV surrounding tachyzoites (*Figure 2F*, top panel, and *Figure 2—video 1*). However, FNR-mCherry was not released from the cyst after A23187 stimulation (*Figure 2F*, middle panel, and *Figure 2—video 3*). As a control to confirm that the FNR-mCherry was indeed secreted into the lumen of the cyst matrix, we treated cysts with trypsin to release bradyzoites. Once the cyst wall was digested, the FNR-mCherry dissipated rapidly, or was digested, confirming that it was present in the matrix of the cyst and not trapped in the parasite (*Figure 2F*, bottom panel, and *Figure 2—video 2*). These data were also confirmed by plotting FNR-mCherry fluorescence intensity changes vs. time for tachyzoites vs. intact or trypsin-treated cysts (*Figure 2G*). These findings demonstrate dampened microneme secretion by bradyzoites, which may explain their incomplete egress.

## Genetically encoded calcium reporter reveals dampened $Ca^{2+}$ responses in bradyzoites

To investigate $Ca^{2+}$ signaling in bradyzoites, we established a dual fluorescent reporter system containing constitutively expressed GCaMP6f and mCherry under the control of bradyzoite stage-specific promoter BAG1 (*Figure 3A*). Using this system, both tachyzoites and bradyzoites express the same levels of GCaMP6f, while only bradyzoites express mCherry, allowing specific monitoring of $Ca^{2+}$ signals in both stages. To confirm the differentiation stage of bradyzoites expressing this dual reporter, we monitored BAG1 and SAG1 expression in bradyzoites induced for different times at alkaline pH by IFA. We observed a gradual increase in the percentage of BAG1-positive and SAG1-negative bradyzoites (mature bradyzoites) from 3 days to 7 days after induction. Based on this criterion, ~70% of parasites were mature bradyzoites in cysts that were induced for 7 days (*Figure 3—figure supplement 1A and B*), and we chose this time point for further studies. Next, we compared the response of BAG1-mCherry GCaMP6f reporter parasites that were grown as tachyzoites to those induced to form bradyzoites by cultivation in HFF cells for 7 days at pH 8.2 in vitro after treatment with $Ca^{2+}$ ionophore A23187. Ionophore treatment induced rapid and high-level increases in GCaMP6f fluorescence in tachyzoites but delayed and much weaker responses in bradyzoites as monitored by time-lapse video microscopy (*Figure 3B*, *Figure 3—video 1*, *Figure 3—video 2*). To rule out an effect of differences in expression level of GCaMP6f during differentiation, we measured fluorescence intensities of GCaMP6f and BAG1-mCherry in different parasites within the same cyst. We observed no correlation between the signals of BAG1-mCherry and the basal expression of GCaMP6f in the absence of ionophore stimulation, indicating that the low responses to ionophore were not due to expression differences in the $Ca^{2+}$-sensitive reporter (*Figure 3—figure supplement 1C and D*). To determine the effect of

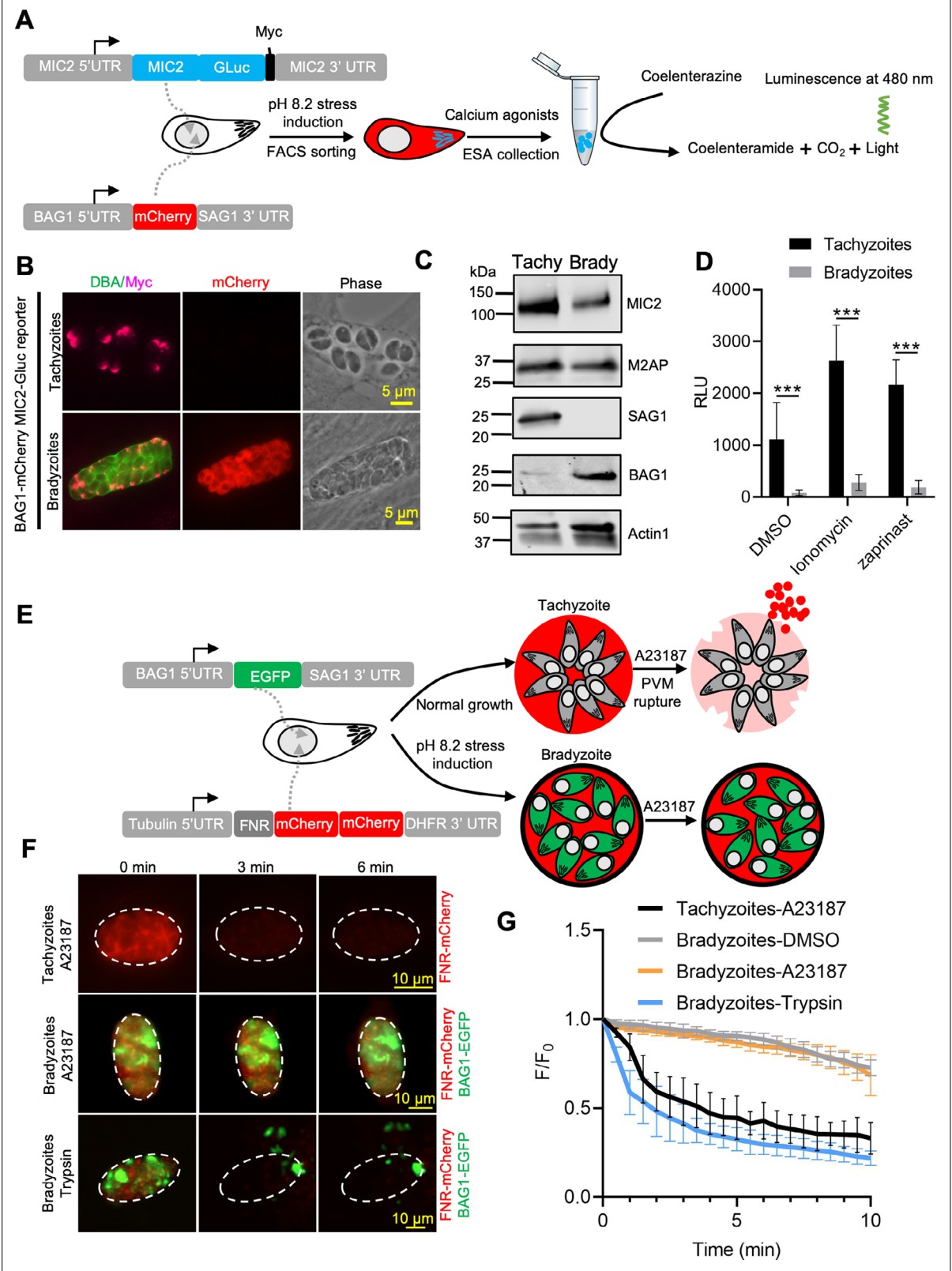

**Figure 2.** Ca²⁺-dependent microneme secretion is significantly dampened in bradyzoites. (**A**) Schematic of bradyzoites MIC2 secretion assay using ME49 BAG1-mCherry MIC2-GLuc bradyzoites, differentiated in vitro by cultivation at pH 8.2 for 7 days, based on fluorescence-activated cell sorting (FACS). (**B**) Immunofluorescence assay (IFA) analysis showing localization of MIC2-Gluc in bradyzoites induced for 7 days at pH 8.2. MIC2-Gluc was stained with anti-Myc antibody, bradyzoites were detected with anti-mCherry, followed by secondary antibodies conjugated with Alexa Fluor dyes, and the cyst wall

*Figure 2 continued on next page*

*Figure 2 continued*

was stained with DBA-FITC. Bar = 5 µm. (**C**) Western blots showing the expression of MIC2-Gluc and M2AP in tachyzoites and bradyzoites (induced for 7 days at pH 8.2, purified by magnetic beads and released from in vitro cysts by 0.25 mg/ml trypsin) of ME49 BAG1-mCherry MIC2-GLuc reporter. αMyc and αM2AP antibodies were used to probe the expression of MIC2-GLuc and M2AP, respectively. SAG1 and BAG1 serve as the stage-specific markers of tachyzoites and bradyzoites, respectively. Actin was used as a loading control. (**D**) ME49 BAG1-mCherry MIC2-Gluc tachyzoites or bradyzoites sorted by FACS and resuspended in extracellular (EC) buffer with $Ca^{2+}$ were stimulated by 0.1% DMSO, ionomycin (1 µM), or zaprinast (500 µM) for 10 min at 37°C. Release of MIC2-GLuc in excretory-secretory antigens (ESA) was determined using a *Gaussia* luciferase assay. RLU indicates relative light units. Means ± SEM of three independent experiments each with three replicates. Multiple Student's *t*-tests, ***p<0.001. (**E**) Schematic illustration of the FNR-mCherry BAG1-EGFP dual fluorescence reporter and leakage of FNR-mCherry from the parasitophorous vacuole (PV) (top) or cyst matrix (bottom) following A23187-induced membrane permeabilization. (**F**) FNR-mCherry leakage was monitored by time-lapse imaging of FNR-mCherry after A23187 (2 µM) treatment. FNR-mCherry BAG1-EGFP tachyzoites cultured under normal condition for 24 hr or bradyzoites induced for 7 days at pH 8.2 were treated with A23187 (2 µM) or 0.25 mg/ml trypsin in EC buffer with $Ca^{2+}$ for 10 min at 37°C. Dash circle indicates the region of interest (ROI) for measurement of fluorescence intensity. Bar = 10 µm. (**G**) FNR-mCherry fluorescence (F) over the initial signal ($F_0$) vs. time from cells treated as in (**F**). Curves are made of data from five independent vacuoles or cysts and shown as means ± SD. Bradyzoites treated with the DMSO group were used to assess photobleaching of mCherry (gray line). DBA, *Dolichos biflorus* agglutinin.

The online version of this article includes the following video and figure supplement(s) for figure 2:

**Source data 1.** Western blotting analysis of ME49 BAG1-mCherry MIC2-GLuc reporter parasites (related to *Figure 2C*).

**Source data 2.** Determining MIC2-Gluc secretion by parasites using *Gaussia* luciferase assay (related to *Figure 2D*).

**Source data 3.** Monitoring leakage of FNR-mCherry from the parasitophorous vacuole (PV) or cyst matrix following A23187-induced membrane permeabilization (related to *Figure 2G*).

**Figure supplement 1.** Validation of ME49 BAG1-mCherry MIC2-GLuc reporter.

**Figure 2—video 1.** A23187-induced permeabilization of the parasitophorous vacuole membrane (PVM) detected by vacuolar leakage of FNR-mCherry secreted by tachyzoites.

https://elifesciences.org/articles/73011/figures#fig2video1

**Figure 2—video 2.** Trypsin-induced disruption of in vitro-differentiated tissue cysts expressing ME49 FNR-mCherry BAG1-EGFP.

https://elifesciences.org/articles/73011/figures#fig2video2

**Figure 2—video 3.** A23187-induced permeabilization of in vitro-differentiated tissue cysts detected by vacuolar FNR-mCherry leakage from ME49 FNR-mCherry BAG1-EGFP bradyzoites.

https://elifesciences.org/articles/73011/figures#fig2video3

---

bradyzoite development on $Ca^{2+}$ signaling, we treated intracellular tachyzoites vs. bradyzoites induced by cultivation in HFF cells at pH 8.2 in vitro for 4–7 days and quantified time of each tachyzoite vacuole or bradyzoite cyst to reach $Ca^{2+}$ peak level after addition of A23187 ionophore by video microscopy. Increasing time of bradyzoites development was associated with progressively longer times to reach peak fluorescence of GCaMP6f (*Figure 3C*). Time-lapse recording of GCaMP6f fluorescence intensity ratio changes ($F/F_0$) showed delayed $Ca^{2+}$ increase and lower fold changes in bradyzoites compared with tachyzoites in response to A23187 stimulation (*Figure 3D*). Zaprinast also elicited slower $Ca^{2+}$ increases and lower fold changes in bradyzoites compared with tachyzoites even in the presence of extracellular $Ca^{2+}$ (*Figure 3E*). To better characterize $Ca^{2+}$ responses of bradyzoites, we performed live video imaging using spinning disc confocal microscopy to distinguish individual bradyzoites within in vitro-differentiated cysts and identify motile bradyzoites within cysts by comparing consecutive images (*Figure 3F*). Motile bradyzoites were also observed to have higher GCaMP6f signals and these typically oscillated over time. In response to $Ca^{2+}$ agonists, intracellular bradyzoites showed reduced percentages of motility compared to tachyzoites (*Figure 3G*). In summary, $Ca^{2+}$ dynamics are delayed and reduced in bradyzoites in response to $Ca^{2+}$ agonists.

## Bradyzoites formed in skeletal muscle cell and within ex vivo cysts show diminished $Ca^{2+}$ responses

To rule out the possibility that alkaline pH stress used for differentiation resulted in lowered $Ca^{2+}$ signals in bradyzoites, we examined $Ca^{2+}$ signaling in bradyzoites within cysts that formed naturally in differentiated C2C12 myocytes. Differentiated myocytes stained positively for skeletal myosin and facilitated the development of bradyzoites, as shown using the bradyzoite stage-specific protein BAG1 (*Figure 4A*). We tested $Ca^{2+}$ responses of bradyzoites formed in muscle cells using the dual fluorescent reporter GCaMP6f BAG1-mCherry parasites in response to A23187 or zaprinast by time-lapse video recording. Time-lapse imaging showed slow increase of GCaMP6f fluorescence in

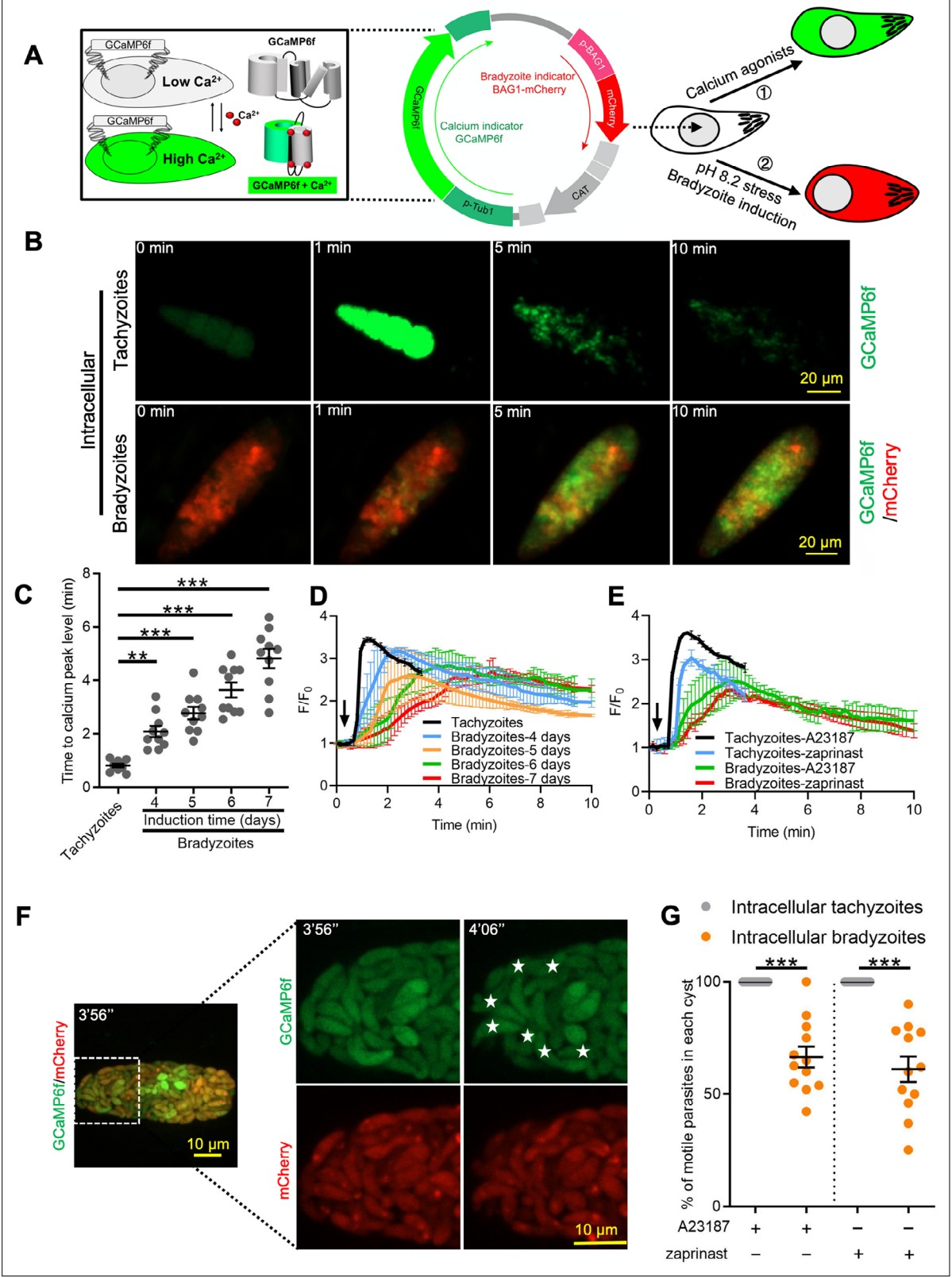

**Figure 3.** Ca²⁺ signaling is dampened during in vitro bradyzoite development induced by alkaline pH. (**A**) Schematic of generation of BAG1-mCherry and GCaMP6f dual fluorescent reporter to monitor Ca²⁺ responses in bradyzoites. (**B**) Time-lapse images of BAG1-mCherry GCaMP6f tachyzoites cultured for 24 hr vs. bradyzoites induced for 7 days at pH 8.2 in response to A23187 (2 μM) in extracellular (EC) buffer with Ca²⁺ for 10 min. Bar = 20 μm. (**C**) Time for reaching Ca²⁺ peak level in response to A23187 (2 μM) for BAG1-mCherry GCaMP6f-expressing tachyzoites and bradyzoites induced at

*Figure 3 continued on next page*

*Figure 3 continued*

pH 8.2. Data points of each group represent 10 cysts or vacuoles. Means ± SD of two independent experiments (n = 10). One-way ANOVA with Dunn's multiple comparison correction test **p<0.01, ***p<0.001. (**D**) Monitoring the relative intensity of GCaMP6f fluorescence fold change (F/F$_0$) vs. time for intracellular tachyzoites and in vitro-induced bradyzoites induced at pH 8.2. Cells were treated with A23187 (2 µM) in EC buffer without Ca$^{2+}$ for 10 min. Curves are the mean fluorescence intensity of five vacuoles or cysts and shown as means ± SD. Arrow indicates time of addition of A23187. (**E**) Monitoring the relative intensity of GCaMP6f fluorescence vs. time for intracellular tachyzoites and in vitro-induced bradyzoites (5 days at pH 8.2). Cells were treated with A23187 (2 µM) or zaprinast (500 µM) in EC buffer with Ca$^{2+}$. Arrow indicates time of addition of agonists. Curves represent the mean data of five independent cysts or vacuoles and are shown as means ± SD. (**F**) Live time-lapse imaging of BAG1-mCherry GCaMP6f bradyzoites induced for 7 days at pH 8.2 in response to A23187 (2 µM) in EC buffer with Ca$^{2+}$. Cells were imaged by spinning disc confocal microscopy after reaching Ca$^{2+}$ peak levels (left panel). Right panel shows its corresponding zoomed-in images. The interval between two continuous images is 10 s, white asterisks in the latter image (4'06") indicate motile bradyzoites by comparison with the former image (3'56"). Bar = 10 µm. (**G**) Motility of parasites within parasitophorous vacuoles (PVs) or cysts was analyzed by time-lapse spinning disc confocal microscopy and tracking of individual parasites for 5 min after reaching Ca$^{2+}$ peak levels in response to A23187 (2 µM) or zaprinast (500 µM) in EC buffer with Ca$^{2+}$. Each data point represents parasites from one vacuole or cyst (n = 10). Data come from two independent experiments. Two-tailed Mann–Whitney test, ***p<0.001. Lines and error bars represent means ± SD of two independent experiments (n = 10).

The online version of this article includes the following video and figure supplement(s) for figure 3:

**Source data 1.** Time of Ca$^{2+}$ dual reporter tachyzoites and bradyzoites with different ages to Ca$^{2+}$ peak level in response to A23187 (related to *Figure 3C*).

**Source data 2.** GCaMP6f fluorescence intensity changes vs. time of GCaMP6f BAG1-mCherry tachyzoites and bradyzoites (related to *Figure 3D*).

**Source data 3.** Zaprinast-induced Ca$^{2+}$ responses in GCaMP6f BAG1-mCherry tachyzoites and bradyzoites (related to *Figure 3E*).

**Source data 4.** Percentage of motile parasites within parasitophorous vacuole or cyst in response to Ca$^{2+}$ agonists (related to *Figure 3G*).

**Figure supplement 1.** Effect of developmental heterology of bradyzoites on GCaMP6f basal signals.

**Figure supplement 1—source data 1.** Developmental heterology of bradyzoites within cyst induced in vitro (related to *Figure 3—figure supplement 1B*).

**Figure supplement 1—source data 2.** Fluorescent intensities of BAG1-mCherry and GCaMP6f of bradyzoites within the same cyst (related to *Figure 3—figure supplement 1D*).

**Figure 3—video 1.** Ca$^{2+}$ response of ME49 BAG1-Cherry GCaMP6f-expressing tachyzoites stimulated by A23187.
https://elifesciences.org/articles/73011/figures#fig3video1

**Figure 3—video 2.** Ca$^{2+}$ response of ME49 BAG1-Cherry GCaMP6f-expressing bradyzoites stimulated by A23187.
https://elifesciences.org/articles/73011/figures#fig3video2

---

response to A23187 in tissue cysts formed in C2C12 myocytes (*Figure 4B*). Both the rate of increase and the maximum amplitude of the GCaMP6f signal were much lower in bradyzoites differentiated in myocytes compared to tachyzoites cultured in undifferentiated myoblasts (*Figure 4C*). The time to reach the peak GCaMP6f fluorescence was also delayed in bradyzoites formed in C2C12 myocytes compared with tachyzoites grown in myoblasts (*Figure 4D*). Bradyzoites cultured in C2C12 myocytes show significantly lower motility in response to A23187 and zaprinast when compared with tachyzoites (*Figure 4E*).

To further examine Ca$^{2+}$ signaling in bradyzoites, we harvested tissue cysts containing BAG1-mCherry GCaMP6f bradyzoites from the brains of chronically infected CD-1 mice and investigated their responses ex vivo. Video microscopy of ex vivo tissue cysts showed slow increases in GCaMP6f fluorescence in response to A23187 or zaprinast (*Figure 4F*). The ratio of GCaMP6f fluorescence changes vs. time (F/F$_0$) from bradyzoites within ex vivo cysts demonstrated lower and slower changes, consistent with lower Ca$^{2+}$ levels, compared with extracellular tachyzoites in response to Ca$^{2+}$ agonists (*Figure 4G*). In comparing the response of extracellular, ex vivo tissue cysts (*Figure 4F and G*) to intracellular cysts formed during infection of C2C12 myocytes (*Figure 4B and C*), it was evident that the extracellular cysts respond somewhat faster, albeit still much slower than tachyzoites. This intermediate level of response was also seen in in vitro-differentiated tissue cyst (produced by cultivation in HFF cells at pH 8.2 for 7 days) that were liberated from HFF cells and tested in vitro (*Figure 4—figure supplement 1*). Next, we measured the percentage of motile and egressed bradyzoites within ex vivo tissue cyst treated with A23187 and zaprinast. Strikingly, no egressed bradyzoites were observed, although all the bradyzoites within ex vivo cysts became motile after stimulation (*Figure 4H*, *Figure 4—video 1*, *Figure 4—video 2*). Taken together, these findings indicate that bradyzoites formed spontaneously

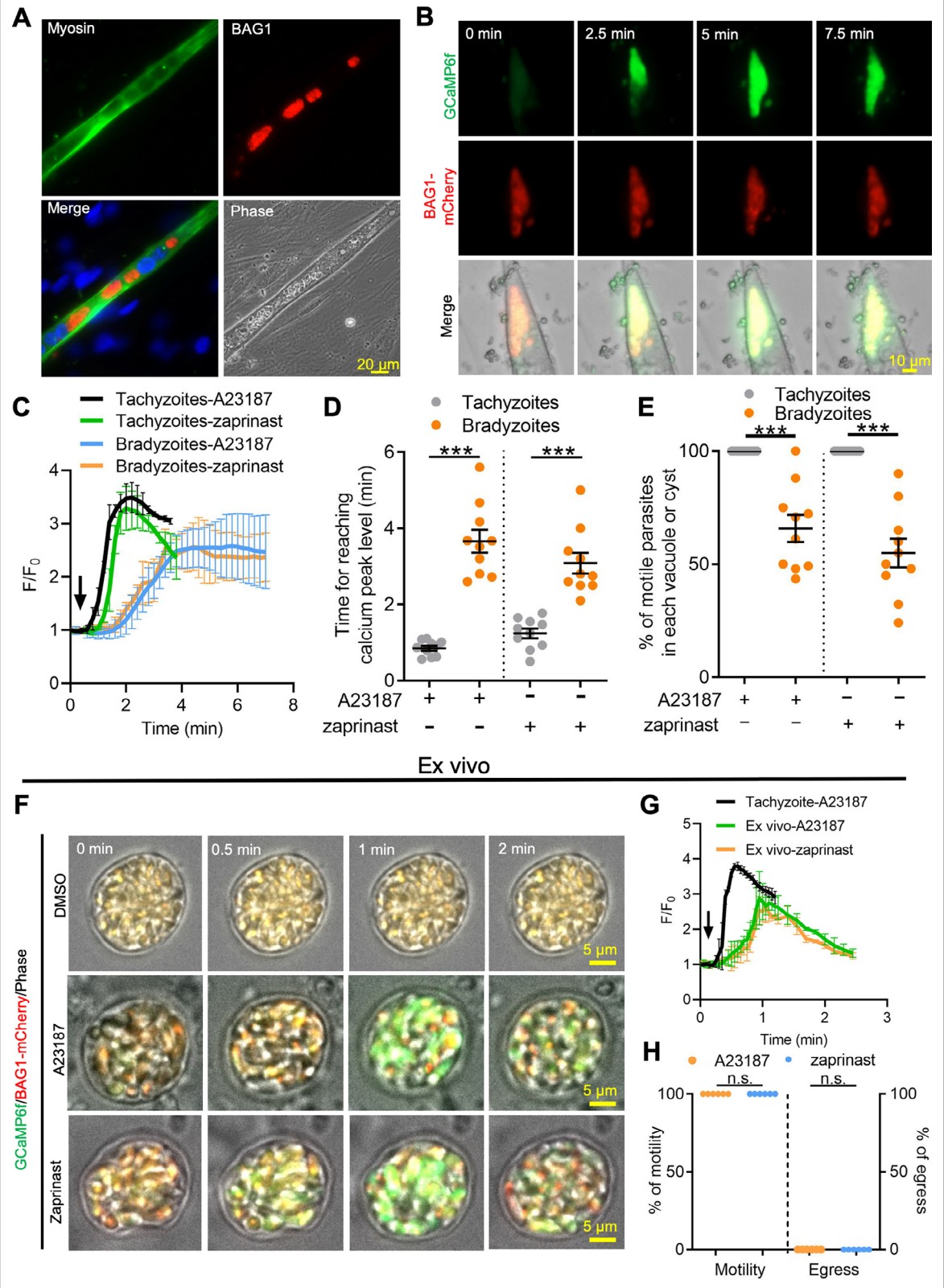

**Figure 4.** Ca$^{2+}$ signaling is dampened in in vitro bradyzoites from spontaneously formed cysts in C2C12 muscle cells and cysts isolated from chronically infected mice. (**A**) Microscopy-based assay for detection of bradyzoites naturally formed after 7 days of culture of the BAG1-mCherry GCaMP6f-expressing dual reporter strain in differentiated C2C12 muscle cells. Anti-myosin antibody was used to confirm the differentiation of C2C12 cells while BAG1 was used to detect bradyzoites followed by secondary antibodies conjugated with Alexa Fluor dyes. Bar = 20 μm. (**B**) Time-lapse recording of

*Figure 4 continued on next page*

*Figure 4 continued*

GCaMP6f fluorescence intensity from cysts of the BAG1-mCherry GCaMP6f strain naturally formed after 7 days culture in C2C12 cells. Cells were treated with A23187 (2 μM) in extracellular (EC) buffer with $Ca^{2+}$. Bar = 10 μm. (**C**) GCaMP6f fluorescence intensity changes vs. time from tachyzoites cultured in undifferentiated myoblasts or cysts naturally formed after 10 days in differentiated C2C12 cells in response to A23187 (2 μM) or zaprinast (500 μM) in EC buffer with $Ca^{2+}$. Curves represent mean data of five independent cysts or vacuoles and are shown as means ± SD. (**D**) Time for reaching $Ca^{2+}$ peak levels in tachyzoites cultured in undifferentiated myoblasts and bradyzoites formed after 10 days culturing in C2C12 cells. Cells were treated with A23187 (2 μM) or zaprinast (500 μM) in EC buffer with $Ca^{2+}$ for 10 min. Data points of each group come from 10 cysts or vacuoles of two independent experiments. Two-tailed unpaired Student's $t$-test, ***$p<0.001$. Lines represent means ± SD of two independent experiments (n = 10). (**E**) Motility of parasites analyzed by time-lapse spinning disc confocal microscopy and tracking of individual parasites for 5 min after reaching $Ca^{2+}$ peak levels in response to A23187 (2 μM) or zaprinast (500 μM) in EC buffer with $Ca^{2+}$. Lines represent means ± SD of two independent experiments (n = 10). Two-tailed Mann–Whitney $t$-test, ***$p<0.001$. (**F**) Monitoring of GCaMP6f fluorescence in response to 0.1% DMSO, A23187 (2 μM), or zaprinast (500 μM) in EC buffer with $Ca^{2+}$ in ex vivo cysts isolated from the brains of mice infected with BAG1-mCherry GCaMP6f reporter parasites. Cysts were harvested at 30 days post infection. Bar = 5 μm. (**G**) GCaMP6f fluorescence intensity changes vs. time within BAG1-mCherry GCaMP6f ex vivo cysts in response to A23187 (2 μM) or zaprinast (500 μM) in EC buffer with $Ca^{2+}$. Curves are the mean data of five independent cysts and are shown as means ± SD. (**H**) Quantitative analysis of motility and egress by bradyzoites from ex vivo cysts isolated from CD-1 mice brain tissues at 30 days post infection. Motility was analyzed by time-lapse microscopy and tracking of individual parasites using time points similar to (**D, E**). Each data point represents percentage of motile or egressed parasites from one cyst (n = 5). Significance was determined by two-tailed Student's $t$-test, n.s., not significant.

The online version of this article includes the following video and figure supplement(s) for figure 4:

**Source data 1.** GCaMP6f fluorescence intensity changes vs. time of GCaMP6f BAG1-mCherry tachyzoites and bradyzoites cultured in C2C12 muscle cells in response to $Ca^{2+}$ agonists (related to *Figure 4C*).

**Source data 2.** Time of tachyzoites and bradyzoites cultured in C2C12 muscle cells to reach GCaMP6f fluorescence peak in response to $Ca^{2+}$ agonists (related to *Figure 4D*).

**Source data 3.** Percentage of motile parasites within parasitophorous vacuole or cyst cultured in C2C12 muscle cells (related to *Figure 4E*).

**Source data 4.** Calcium responses of ex vivo cysts in response to $Ca^{2+}$ agonists (related to *Figure 4G*).

**Source data 5.** Motility and egress of bradyzoites within ex vivo cysts isolated from chronically infected mice in response to A23187 (related to *Figure 4H*).

**Figure supplement 1.** Calcium responses by extracellular tachyzoites and in vitro-produced tissue cysts.

**Figure supplement 1—source data 1.** Calcium responses by extracellular tachyzoites and in vitro-produced tissue cysts (related to *Figure 4—figure supplement 1A*).

**Figure 4—video 1.** $Ca^{2+}$ response of ME49 BAG1-mCherry GCaMP6f cysts isolated from chronically infected mouse brains and treated in vitro with DMSO.

https://elifesciences.org/articles/73011/figures#fig4video1

**Figure 4—video 2.** Calcium response of ME49 BAG1-mCherry GCaMP6f cysts isolated from chronically infected mouse brains and treated in vitro with A23187.

https://elifesciences.org/articles/73011/figures#fig4video2

in muscle myocytes and within ex vivo cysts from chronically infected mice display dampened $Ca^{2+}$ dynamics when treated with $Ca^{2+}$ agonists.

## Bradyzoites store less $Ca^{2+}$ in ER and acidocalcisome

The cyst wall surrounding bradyzoites may restrict access to $Ca^{2+}$ agonists and hence dampen signals from GCaMP6f in response to $Ca^{2+}$ agonists in the studies described above. To test this possibility, we monitored GCaMP6f fluorescence changes in extracellular bradyzoites vs. tachyzoites of the BAG1-mCherry GCaMP6f strain by live imaging. Bradyzoites were induced by cultivation in HFF cells at pH 8.2 for 7 days and liberated from cysts by trypsin treatment, followed by washing and resuspension for analysis. We also observed slower increases in GCaMP6f fluorescence intensity in bradyzoites (*Figure 5—video 2*) compared with tachyzoites (*Figure 5—video 1*) in response to A23187 (*Figure 5A*). Quantitative analysis of $Ca^{2+}$ fluorescence changes ($F/F_0$) after stimulation by A23187 and zaprinast showed slower $Ca^{2+}$ responses in extracellular bradyzoites when compared to tachyzoites (*Figure 5B*). To confirm that extracellular bradyzoites were viable after liberation from in vitro-cultured cysts by trypsin treatment, we utilized SYTOX Red, which is a DNA dye excluded by intact membranes of viable cells. In contrast to bradyzoites that were formaldehyde-fixed as a positive control, extracellular bradyzoites were not stained by SYTOX after the liberation from in vitro cysts (*Figure 5C*), indicating that they were still viable after trypsin treatment.

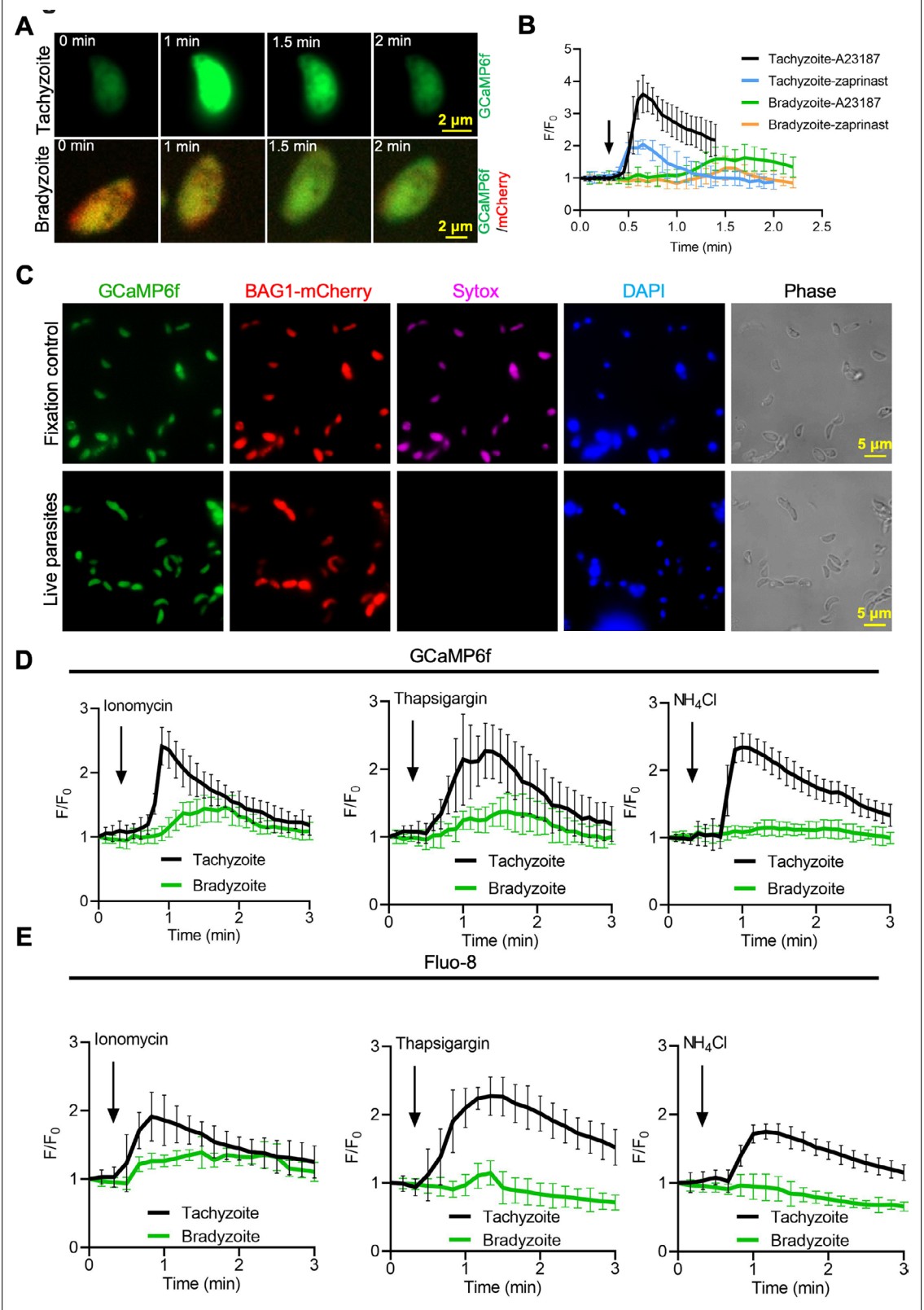

**Figure 5.** Bradyzoites have lower $Ca^{2+}$ stores and reduced responses to agonists compared to tachyzoites. (**A**) Live imaging of extracellular BAG1-mCherry GCaMP6f dual fluorescent reporter tachyzoites and bradyzoites induced for 7 days at pH 8.2 in response to A23187 (2 μM) in extracellular (EC) buffer with $Ca^{2+}$. Bar = 2 μm. (**B**) Fluorescence recording of increased GCaMP6f fluorescence with $Ca^{2+}$ increase in response to A23187 (2 μM) or zaprinast (500 μM) in EC buffer with $Ca^{2+}$ for extracellular tachyzoites and bradyzoites. Arrow indicates the addition of $Ca^{2+}$ agonists. Each curve is the

*Figure 5 continued on next page*

Figure 5 continued

mean of 10 individual parasites and shown as means ± SD. (**C**) BAG1-mCherry GCaMP6f reporter live bradyzoites were stained by SYTOX far red to detect dead cells and DAPI 30 min after liberation from cysts. Formaldehyde-fixed bradyzoites serve as positive control. Bar = 5 μm. (**D**) GCaMP6f fluorescence intensity vs. time for extracellular BAG1-mCherry GCaMP6f dual reporter parasites in response to 1 μM ionomycin, 1 μM thapsigargin, or 10 mM NH$_4$Cl in EC buffer without Ca$^{2+}$. Arrow indicates the addition of agonist. Each curve is the mean of 10 individual parasites and shown as means ± SD. (**E**) Fluorescence intensities change fold vs. time of extracellular BAG1-mCherry expressing bradyzoites loaded with 500 nM Fluo-8 AM after addition of 1 μM ionomycin, 1 μM thapsigargin, or 10 mM NH$_4$Cl in EC buffer without Ca$^{2+}$. Arrow indicates the addition of agonist. Each curve is the mean of 10 individual parasites and shown as means ± SD.

The online version of this article includes the following video and figure supplement(s) for figure 5:

**Source data 1.** Calcium responses of extracellular single tachyzoite and bradyzoite stimulated by Ca$^{2+}$ agonists (related to *Figure 5B*).

**Source data 2.** Calcium responses of extracellular GCaMP6f BAG1-mCherry tachyzoites and bradyzoites treated with ionomycin, thapsigargin, and NH$_4$Cl (related to *Figure 5D*).

**Source data 3.** Calcium responses of extracellular Fluo-8-loaded BAG1-mCherry tachyzoites and bradyzoites treated with ionomycin, thapsigargin, and NH$_4$Cl (related to *Figure 5E*).

**Figure 5—video 1.** Ca$^{2+}$ response of extracellular ME49 BAG1-mCherry GCaMP6f tachyzoite in response to A23187.
https://elifesciences.org/articles/73011/figures#fig5video1

**Figure 5—video 2.** Ca$^{2+}$ response of extracellular ME49 BAG1-mCherry GCaMP6f bradyzoite in response to A23187.
https://elifesciences.org/articles/73011/figures#fig5video2

We hypothesized that bradyzoites might have dampened GCaMP6f responses because they fail to release Ca$^{2+}$ from intracellular stores. We tested Ca$^{2+}$ responses of BAG1-mCherry and GCaMP6f-expressing bradyzoites and tachyzoites treated with ionomycin, which releases Ca$^{2+}$ mainly from the ER (*Beeler et al., 1979*), thapsigargin, which inhibits SERCA-type Ca$^{2+}$-ATPase causing an increase of cytosolic Ca$^{2+}$ due to uncompensated leakage from the ER (*Thastrup et al., 1990*), and NH$_4$Cl, an alkalizing reagent that releases Ca$^{2+}$ from acidic stores like acidocalcisomes (*Moreno and Zhong, 1996*). Both ionomycin and thapsigargin induced delayed and lower amplitude changes in GCaMP6f fluorescence in bradyzoites vs. tachyzoites as shown by plotting fluorescence intensity fold changes (F/F$_0$) vs. time (*Figure 5D*), indicative of lower ER-stored Ca$^{2+}$. In contrast, bradyzoites treated with NH$_4$Cl showed no meaningful change in GCaMP6f fluorescence, suggesting that they lack mobilizable acidic Ca$^{2+}$ (*Figure 5D*). To rule out the possibility that the Ca$^{2+}$ indicator GCaMP6f is less sensitive in bradyzoites due to some intrinsic defect, we loaded BAG1-mCherry-expressing tachyzoite or bradyzoites with the Ca$^{2+}$-sensitive vital dye Fluo-8 AM and used these cells for imaging. Fluo-8 AM-labeled bradyzoites displayed dampened Ca$^{2+}$ signaling after stimulation by ionomycin, thapsigargin, or NH$_4$Cl, relative to tachyzoites that responded normally (*Figure 5E*). Collectively, these findings indicate that bradyzoites are less able to mobilize Ca$^{2+}$ from the ER and acidic stores in response to agonists.

## Ratiometric sensor reveals reduced basal levels of Ca$^{2+}$ and dynamics in bradyzoites

To more precisely compare Ca$^{2+}$ levels in tachyzoites and bradyzoites, we constructed a ratiometric fluorescence reporter by coexpression of GCaMP6f with blue fluorescent protein (BFP) mTagBFP2 linked by a P2A split peptide (*Figure 6A*, *Figure 6—figure supplement 1A*, *Figure 6—figure supplement 1B*). Because both proteins are coexpressed from the same promoter, the mTagBFP2 serves as a control for expression level as mTagBFP2 is nonresponsive to Ca$^{2+}$ levels (*Cranfill et al., 2016*). Live fluorescence microscopy showed simultaneous expression of GCaMP6f and mTagBFP2 in tachyzoites, and additionally mCherry in bradyzoites (*Figure 6B*). Equal expression of GCaMP6f (His tag) and mTagBFP2, as well as separation of tachyzoites and bradyzoite populations (detected with SAG1 and BAG1, respectively), was validated by western blotting (*Figure 6C*). To compare Ca$^{2+}$ basal levels, we quantified the fluorescence intensity ratio F$_{GCaMP6f}$/F$_{mTagBFP2}$ of intracellular and extracellular tachyzoites and bradyzoites in EC buffer with or without Ca$^{2+}$. We observed significant reductions in the fluorescence intensity ratio of both intracellular and extracellular bradyzoites relative to tachyzoites (*Figure 6D*), indicative of lower resting Ca$^{2+}$ levels in bradyzoites. We next compared Ca$^{2+}$ dynamics of intracellular tachyzoites and bradyzoites in response to Ca$^{2+}$ agonists ionomycin, NH$_4$Cl, and thapsigargin. Changes in the fluorescence of GCaMP6f were much slower and of lower amplitude

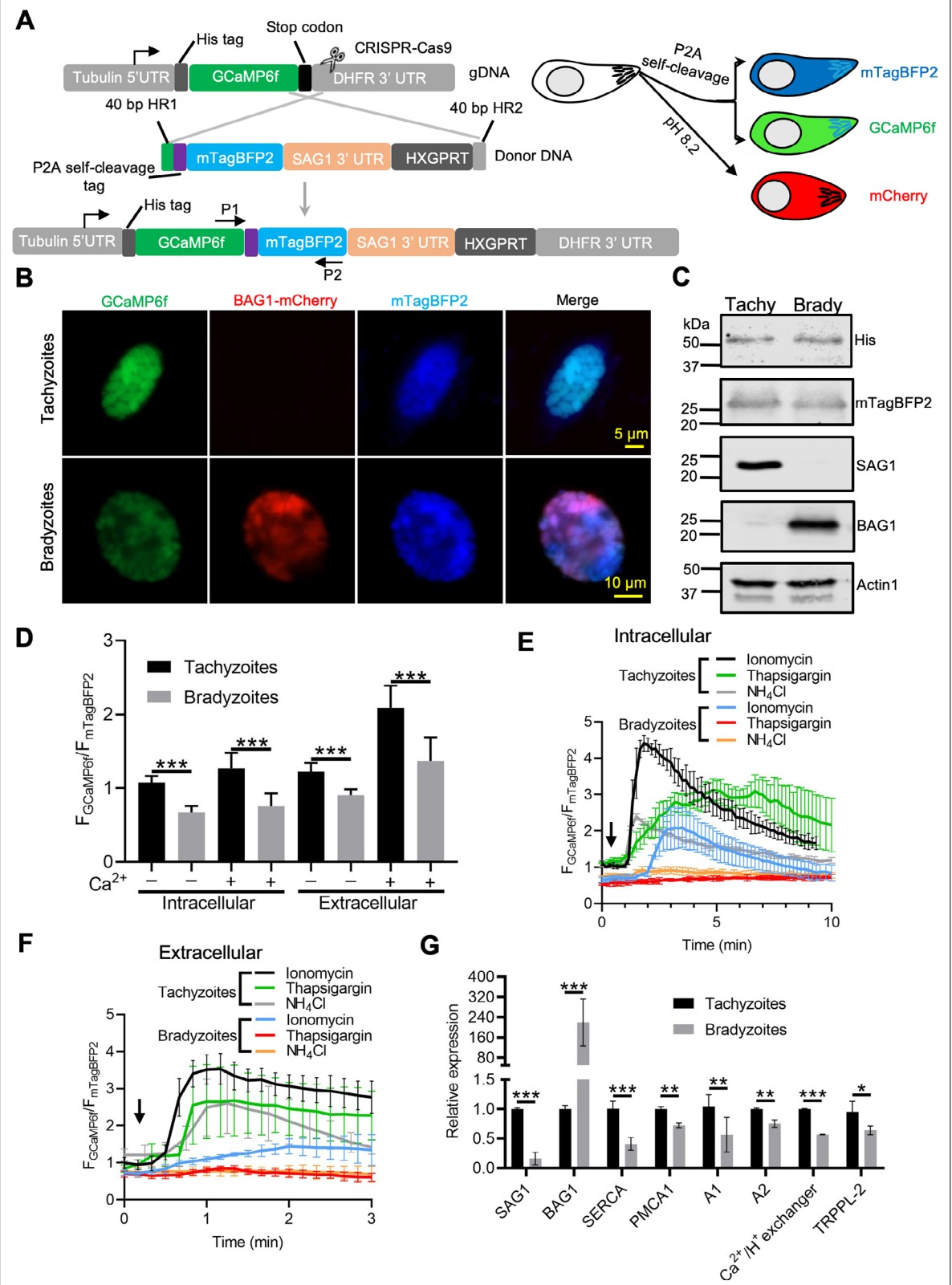

**Figure 6.** Ratiometric Ca²⁺ imaging of bradyzoites reveals lower levels of resting Ca²⁺ and reduced response to Ca²⁺ ionophores compared to tachyzoites. (**A**) Schematic diagram of generation of a ratiometric Ca²⁺ reporter containing GCaMP6f fused with by a peptide P2A and blue fluorescence indicator mTagBFP2 in the background of BAG1-mCherry reporter strain. P1 and P2 are primes used for the diagnostic PCR to confirm the integration of P2A-mTagBFP2 into the C-terminal of GCaMP6f. (**B**) Fluorescence microscopy imaging of the intracellular ratiometric indicator expressed by tachyzoites

*Figure 6 continued on next page*

*Figure 6 continued*

cultured for 24 hr vs. bradyzoites induced for 7 days at pH 8.2 culture in extracellular (EC) buffer without $Ca^{2+}$. Bar = 10 µm. (**C**) Western blots showing GCaMP6f and mTagBFP2 produced from the ratiometric reporter expressed by tachyzoites and bradyzoites. αHis and αtRFP antibodies were used to probe the expression of GCaMP6f and mTagBFP2, respectively. SAG1 and BAG1 serve as the stage-specific marker of tachyzoites and bradyzoites, respectively. Actin functions as loading control. (**D**) Quantification of basal $Ca^{2+}$ levels normalized by comparison of GCaMP6f to mTagBFP2 fluorescence intensity ratios of intracellular and extracellular tachyzoites or bradyzoites that were induced by culture for 7 days at pH 8.2. For extracellular parasites, tachyzoites were liberated mechanically and bradyzoites were liberated by trypsin treatment. Parasites within intact cells or extracellular parasites were incubated in EC buffer with or without $Ca^{2+}$ for 10 min before imaging. Data represent mean values from two independent experiments with 10 total vacuoles or cysts for each treatment. Two-tailed unpaired Student's *t*-test, ***p<0.001. (**E**) Monitoring of GCaMP6f/ mTagBFP2 fluorescence intensity ratio vs. time for intracellular tachyzoites and in vitro-induced bradyzoites that were induced by culture for 7 days at pH 8.2. Each kinetic curve represents mean data of five individual vacuoles or cysts and is shown as means ± SD. (**F**) For extracellular parasites, tachyzoites were liberated mechanically and bradyzoites were liberated by trypsin treatment. Parasites were incubated in EC buffer without $Ca^{2+}$ for 10 min, and responses were measured to ionomycin (1 µM), thapsigargin (1 µM), or 10 mM $NH_4Cl$. Arrow indicates time of addition of agonists. Each kinetic curve represents mean data of 10 individual parasites and is shown as means ± SD. (**G**) Gene expression levels in tachyzoites and bradyzoites induced for 7 days at pH 8.2. mRNA levels were measured using RT-PCR and expressed relative to the housekeeping transcript for actin. SAG1 and BAG1 were used to monitor tachyzoites and bradyzoites, respectively. Data represent the means ± SD of two independent assays containing triplicate samples each. Multiple Student's *t*-tests, **p<0.01, ***p<0.001.

The online version of this article includes the following figure supplement(s) for figure 6:

**Source data 1.** Western blotting analysis of ME49 GCaMP6f-P2A-mTagBFP2 BAG1-mCherry ratiometric reporter parasites (related to *Figure 6C*).

**Source data 2.** Comparison of $Ca^{2+}$ calcium basal levels between tachyzoites and bradyzoites using ratiometric reporter (related to *Figure 6D*).

**Source data 3.** Calcium responses of intracellular GCaMP6f ratiometric reporter tachyzoites and bradyzoites treated with different $Ca^{2+}$ agonists (related to *Figure 6E*).

**Source data 4.** Calcium responses of extracellular GCaMP6f ratiometric reporter tachyzoites and bradyzoites treated with different $Ca^{2+}$ agonists (related to *Figure 6F*).

**Source data 5.** Comparison of mRNA expression levels of genes encoding calcium-associated channels and pumps in tachyzoites and bradyzoites by qPCR (related to *Figure 6G*).

**Figure supplement 1.** Identification of ME49 GCaMP6f-P2A-mTagBFP2 BAG1-mCherry ratiometric reporter by PCR and immunofluorescence assay (IFA).

**Figure supplement 1—source data 1.** Uncropped DNA gel for PCR identification of ratiometric reporter clones (related to *Figure 6—figure supplement 1*).

in bradyzoites relative to tachyzoites (*Figure 6E*). We also observed lower resting $Ca^{2+}$ and peak levels in extracellular bradyzoites compared to tachyzoites (*Figure 6F*), indicating lower activity or expression of cytoplasmic influx mechanisms like the PM $Ca^{2+}$ entry or ER $Ca^{2+}$ release channels. To understand the molecular basis for the reduced stored $Ca^{2+}$ and responses in bradyzoites, we performed real-time PCR to compare mRNA expression levels of several $Ca^{2+}$ transporters and channels. Included in this list are TgSERCA (*Nagamune et al., 2007*), which is the molecular target of thapsigargin and transfers $Ca^{2+}$ from the cytosol of parasites to ER, TgA1 (*Luo et al., 2004*), which plays important roles in the accumulation of $Ca^{2+}$ in the acidocalcisome and other acidic stores, TgTRPPL-2 (*Márquez-Nogueras et al., 2021*), which is a TRP channel key for $Ca^{2+}$ influx into the cytosol, and additional $Ca^{2+}$-related proteins, such as TgPMCA1, TgA2, and the $Ca^{2+}/H^+$ exchanger (*Nagamune et al., 2008*). We observed significant reduction in the relative expression level of TgSERCA, TgA1, TgPMCA1, TgA2, $Ca^{2+}/H^+$ exchanger, and TgTRPPL-2 in bradyzoites compared to tachyzoites (*Figure 6G*). Taken together, these findings indicate that bradyzoites have lower levels of stored $Ca^{2+}$, which is associated with the overall downregulation of $Ca^{2+}$-related pumps and channels.

## Calcium signaling plays a critical role in gliding motility of bradyzoites

To test whether dampened $Ca^{2+}$ signaling would still be sufficient to drive gliding motility of bradyzoites, we treated BAG1-mCherry GCaMP6f-expressing cysts cultured in vitro with trypsin to liberate bradyzoites (*Figure 7A*). There were no obvious changes in the $Ca^{2+}$ levels nor motility during trypsin treatment and release of bradyzoites that were monitored for 5 min (*Figure 7B*, *Figure 7—video 1*). We then extended the trypsin treatment to 10 min to assure complete digestion of the cysts and to allow bradyzoites to settle before imaging by time-lapse microscopy. Again, we observed that the majority of bradyzoites failed to show enhanced GCaMP fluorescence or initiate gliding motility. However, the small number of bradyzoites that did undergo gliding displayed patterns that were highly

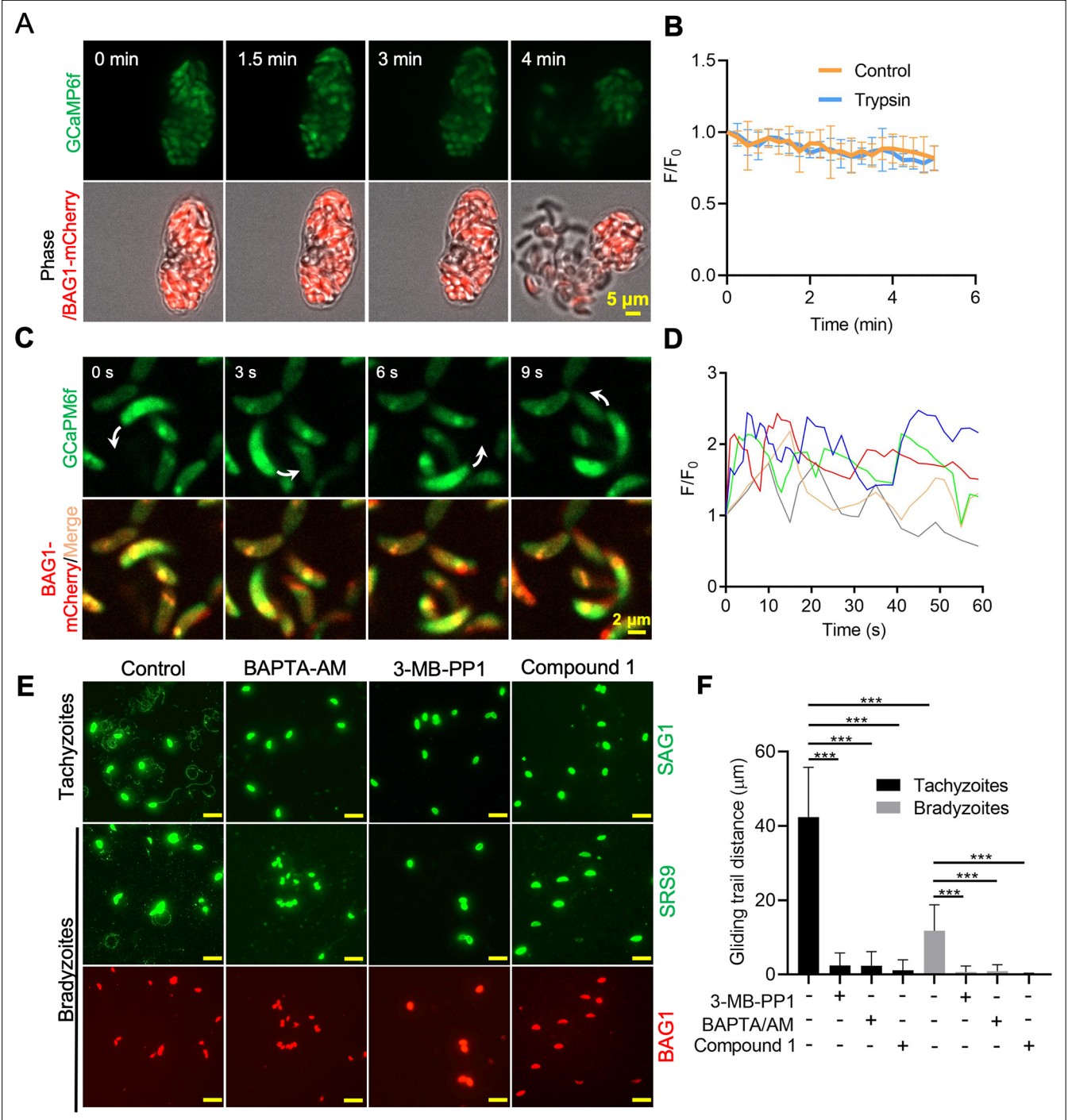

**Figure 7.** Ca²⁺ signaling governs gliding motility of bradyzoites. (**A**) Time-lapse microscopy recording of GCaMP6f BAG1-mCherry bradyzoites induced for 7 days at pH 8.2. Cells were imaged during the digestion by 0.25 mg/ml trypsin for 5 min in extracellular (EC) buffer with 1.8 mM Ca²⁺. Bar = 5 µm. (**B**) GCaMP6f fluorescence change ratio vs. time of BAG1-mCherry GCaMP6f bradyzoites induced for 7 days at pH 8.2 treated with or without trypsin. Curves represent mean data from five independent cysts. (**C**) Spinning disc confocal microscopy monitoring circular gliding motility of bradyzoites liberated by 0.25 mg/ml trypsin for 10 min from cysts induced for 7 days at pH 8.2. Arrow shows the direction of gliding motility by one bradyzoite. Bar = 5 µm. (**D**) Ca²⁺ kinetics of bradyzoites undergoing gliding motility after liberation from cysts induced for 7 days at pH 8.2. The graph shows fluctuated Ca²⁺ kinetics of five independent single bradyzoites. (**E**) Indirect immunofluorescence microscopy showing the trails of parasites during gliding motility. Parasites were treated with DMSO (control), 5 µM 3-MB-PP1, 25 µM BAPTA-AM, and 4 µM compound 1. Anti-SAG1 mAb DG52 and rabbit polyclonal anti-SRS9 antibodies followed by secondary antibodies conjugated to goat anti-mouse IgG Alexa 488 were used to stain the gliding trails of tachyzoites and bradyzoites, respectively. Anti-BAG1 followed by goat anti-rabbit IgG conjugated of Alexa 568 served as marker of bradyzoites. Bar = 10 µm.

*Figure 7 continued on next page*

*Figure 7 continued*

(**F**) Quantification of trails from gliding motility of tachyzoites and bradyzoites treated with DMSO (control), 5 µM 3-MB-PP1, 25 µM BAPTA-AM, and 4 µM compound 1. Data represented as means ± SEM (n = 20 replicates combined from n = 3 independent experiments). Kruskal–Wallis test with Dun's multiple comparison correction ***p<0.001.

The online version of this article includes the following video and figure supplement(s) for figure 7:

**Source data 1.** Calcium responses of GCaMP6f BAG1-mCherry reporter bradyzoites during liberation from in vitro-induced cyst by trypsin (related to *Figure 7B*).

**Source data 2.** Calcium fluctuation of extracellular single bradyzoites (related to *Figure 7D*).

**Source data 3.** Effects of inhibition of calcium signaling pathway on the gliding motility of tachyzoites and bradyzoites (related to *Figure 7F*).

**Figure supplement 1.** Effects of trypsin treatment on tachyzoites $Ca^{2+}$ pools.

**Figure supplement 1—source data 1.** Effects of trypsin treatment on tachyzoites $Ca^{2+}$ pools (related to *Figure 7—figure supplement 1A*).

**Figure 7—video 1.** Trypsin-induced liberation of ME49 BAG1-mCherry GCaMP6f bradyzoites from in vitro-cultured cysts.

https://elifesciences.org/articles/73011/figures#fig7video1

**Figure 7—video 2.** Gliding motility of ME49 BAG1-mCherry GCaMP6f bradyzoites released from in vitro cysts.

https://elifesciences.org/articles/73011/figures#fig7video2

reminiscent of tachyzoite motility including circular gliding (*Figure 7C*, *Figure 7—video 2*). Similar to previous descriptions of oscillating $Ca^{2+}$ patterns in gliding tachyzoites (*Lovett and Sibley, 2003*), we observed fluctuations of GCaMP6f fluorescence intensities in individual extracellular bradyzoites that exhibited gliding motility (*Figure 7D*). To evaluate the potential effect of trypsin treatment on $Ca^{2+}$ levels, we incubated extracellular GCaMP6f-expressing tachyzoites with trypsin and observed no difference in the $Ca^{2+}$ responses induced by ionomycin (*Figure 7—figure supplement 1A*), ruling out the possibility that the low $Ca^{2+}$ response in bradyzoites was caused by trypsin treatment.

To further characterize the role of $Ca^{2+}$ signaling in bradyzoite motility, we treated cells with the $Ca^{2+}$ chelator BAPTA-AM, the PKG inhibitor compound 1, and the CDPK1 inhibitor 3-MB-PP1 to block $Ca^{2+}$ signaling in bradyzoites. All these inhibitors significantly impaired gliding motility of tachyzoites and bradyzoites (*Figure 7E and F*), confirming a key role of $Ca^{2+}$ signaling in parasite motility. Bradyzoites displayed shorter gliding distance compared with tachyzoites as determined by measurements of trail lengths detected with SAG1 (tachyzoite) or SRS9 (bradyzoites) (*Figure 7F*). These two surface markers are both members of the cysteine-rich SRS family that are tethered to the surface membrane by a GPI anchor, thus they represent comparable reporters for each stage. In summary, despite having dampened $Ca^{2+}$ stores and reduced responses to agonist when intracellular, extracellular bradyzoites require $Ca^{2+}$ signaling to activate gliding motility.

## Accumulation of $Ca^{2+}$ stores and ATP synergistically activates gliding motility by bradyzoites

Following reactivation of tissue cysts, we hypothesized that bradyzoites must replenish their $Ca^{2+}$ and energy stores to meet the demands of cell to cell transmission. To test this idea, we released bradyzoites using trypsin treatment and then treated extracellular bradyzoites with EC buffer with or without $Ca^{2+}$ (1.8 mM) and with or without glucose (5.6 mM) for different times and stimulated the $Ca^{2+}$ responses using ionomycin. Quantitative analysis of $Ca^{2+}$ fluorescence changes ($F/F_0$) showed that bradyzoites substantially recovered stored $Ca^{2+}$ in the presence of exogenous $Ca^{2+}$ and glucose for 1 hr compared to 10 min (*Figure 8A and B*). A more modest recovery was observed in the presence of $Ca^{2+}$ but absence of glucose (*Figure 8B*). We also monitored the effect of recovery of $Ca^{2+}$ pools on microneme secretion by bradyzoites and found that bradyzoites secreted more MIC2-GLuc after being treated with exogenous $Ca^{2+}$ and glucose for 1 hr compared to 10 min (*Figure 8C*). Next, we investigated the effect of exogenous $Ca^{2+}$ and glucose on gliding motility by bradyzoites. We used time-lapse video microscopy to determine the percentage of extracellular bradyzoites undergoing twirling, circular and helical motility after incubation in EC buffer ± $Ca^{2+}$ and glucose for 10 min vs. 1 hr. Quantitative analysis showed that bradyzoites underwent all forms of gliding motility and substantially recovered gliding motility after incubation with EC buffer containing both $Ca^{2+}$ and glucose for 1 hr, while very few bradyzoites were able to glide following incubation with exogenous $Ca^{2+}$ or glucose alone (*Figure 8D*).

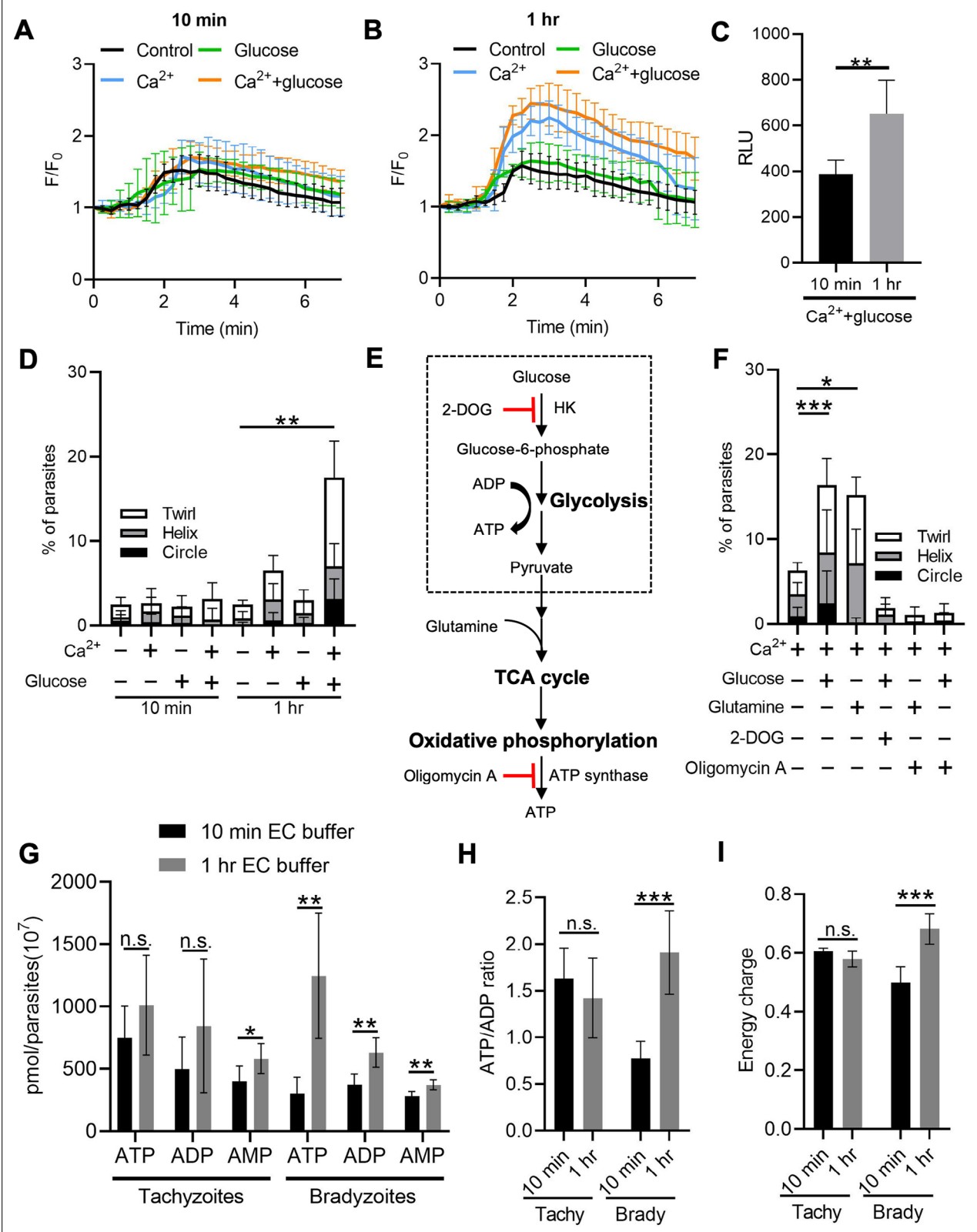

**Figure 8.** Exogenous $Ca^{2+}$ and glucose collectively contributes to bradyzoites gliding motility via refilling $Ca^{2+}$ pools and increasing ATP production. (**A, B**) Monitoring the relative intensity of GCaMP6f fluorescence fold change ($F/F_0$) vs. time from extracellular bradyzoites treated with 1 μM ionomycin. Bradyzoites induced for 7 days at pH 8.2 were released from in vitro cysts by 0.25 mg/ml trypsin and pre-incubated in extracellular (EC) buffer ±1.8 mM $Ca^{2+}$ and/or ± 5.6 mM glucose for 10 min (**A**) or 1 hr (**B**) before measurements. Each kinetic curve represents mean data of 10 individual extracellular

*Figure 8 continued on next page*

*Figure 8 continued*

parasites and is shown as means ± SD. Arrow indicates the addition of 1 µM ionomycin. Control refers to the absence of Ca²⁺ and glucose. (**C**) ME49 BAG1-mCherry MIC2-Gluc bradyzoites were induced for 7 days at pH 8.2, purified by magnetic beads, released from in vitro cysts by 0.25 mg/ml trypsin and incubated in EC buffer containing 1.8 mM Ca²⁺ and 5.6 mM glucose for 10 min or 1 hr, followed by stimulation with ionomycin (1 µM) for 10 min. Release of MIC2-GLuc in ESA was determined using a *Gaussia* luciferase assay. RLU indicates relative light units. Means ± SD of two independent experiments (n = 6). Two-tailed Mann–Whitney test, **p<0.01. (**D**) Percentage of extracellular parasites undergoing different forms of gliding motility as determined from time-lapse video microscopy. Bradyzoites induced for 7 days at pH 8.2 were treated in EC buffer ± 1.8 mM Ca²⁺ and/or ±5.6 mM glucose for 10 min or 1 hr before measurements. Means ± SD of two independent experiments with six replicates each. Kruskal–Wallis test with Dunn's multiple comparison correction test **p<0.01 for comparison between –Ca²⁺/– glucose and +Ca²⁺/+ glucose. All other groups were not significantly different from the negative control. (**E**) Schematic illustration of the mechanism of 2-deoxyglucose (2-DOG) and oligomycin A in inhibiting ATP production. (**F**) Percentage of bradyzoites with different forms of gliding motility determined by time-lapse video microscopy. Bradyzoites induced for 7 days at pH 8.2 were treated in EC buffer (1.8 mM Ca²⁺) ± 5.6 mM glucose, 5.6 mM glutamine, 50 mM 2-DOG, or 20 µM oligomycin A for 1 hr before measurements. Means ± SD of two independent experiments with six replicates each. Kruskal–Wallis test with Dunn's multiple comparison correction test *p<0.05, ***p<0.001. (**G–I**) High-performance liquid chromatography UV (HPLC-UV) analysis of adenosine triphosphate (ATP), adenosine diphosphate (ADP), and adenosine monophosphate (AMP) levels in extracellular tachyzoites and bradyzoites incubated with EC buffer containing 1.8 mM Ca²⁺ and 5.6 mM glucose for 10 min or 1 hr. Bradyzoites induced for 7 days at pH 8.2 were purified by magnetic beads and released from in vitro cysts by 0.25 mg/ml trypsin. Data from two independent experiments with six technical replicates. (**G**) Concentrations of ATP, ADP, and AMP in extracellular tachyzoites and bradyzoites represented as means ± SD. Multiple Student's *t*-tests, n.s., not significant, *p<0.05, **p<0.01. (**H**) ATP/ADP ratios in extracellular tachyzoites and bradyzoites represented as means ± SD. Multiple Student's *t*-tests, n.s., not significant, ***p<0.001. (**I**) Energy charge of extracellular bradyzoites calculated as [ATP] + 0.5 × [ADP]/[ATP] + [ADP] + [AMP] represented as means ± SD. Multiple Student's *t*-tests, n.s., not significant, ***p<0.001.

The online version of this article includes the following figure supplement(s) for figure 8:

**Source data 1.** Calcium fluorescence fold change of extracellular BAG1-mCherry GCaMP6f dual reporter bradyzoites treated with or without Ca²⁺ or glucose for 10 min in response to ionomycin (related to *Figure 8A*).

**Source data 2.** Calcium fluorescence fold change of extracellular BAG1-mCherry GCaMP6f dual reporter bradyzoites treated with or without Ca²⁺ or glucose for 1 hr in response to ionomycin (related to *Figure 8B*).

**Source data 3.** Recovery of microneme secretion by bradyzoites in extracellular (EC) buffer containing Ca²⁺ and glucose (related to *Figure 8C*).

**Source data 4.** Percentage of gliding motility of bradyzoites treated with or without glucose or Ca²⁺ for 10 min or 1 hr (related to *Figure 8D*).

**Source data 5.** Effects of inhibition of adenosine triphosphate (ATP) production pathways on the gliding motility of bradyzoites (related to *Figure 8F*).

**Source data 6.** High-performance liquid chromatography (HPLC) analysis of adenosine triphosphate (ATP), adenosine diphosphate (ADP), and adenosine monophosphate (AMP) levels in bradyzoites treated with glucose and calcium containing extracellular (EC) buffer for 10 min or 1 hr (related to *Figure 8G*).

**Source data 7.** Adenosine triphosphate (ATP)/adenosine diphosphate (ADP) ratio of bradyzoites treated with glucose and Ca²⁺ containing extracellular (EC) buffer for 10 min or 1 hr (related to *Figure 8H*).

**Source data 8.** Energy charge of bradyzoites treated with glucose and Ca²⁺ containing extracellular (EC) buffer for 10 min or 1 hr (related to *Figure 8I*).

**Figure supplement 1.** Establishment of high-performance liquid chromatography UV (HPLC-UV) analysis of adenosine triphosphate (ATP), adenosine diphosphate (ADP), and adenosine monophosphate (AMP) levels in parasites.

**Figure supplement 1—source data 1.** Chromatograms from high-performance liquid chromatography (HPLC) analysis of adenosine triphosphate (ATP), adenosine diphosphate (ADP), and adenosine monophosphate (AMP) contents in parasites (related to *Figure 8—figure supplement 1A–C* ).

We reasoned that exogenous glucose could be utilized by parasites to produce ATP via glycolysis or oxidative phosphorylation to maintain a variety of cellular functions. To investigate the ATP source for supporting gliding motility, we treated exogenous bradyzoites in EC buffer containing Ca²⁺ (1.8 mM) with glucose to support glycolysis vs. the glucose analogue 2-deoxy-D-glucose (2-DOG) to block glycolysis (*Figure 8E*). Alternatively, similar preparations of bradyzoites were incubated with glutamine to provide substrates for the tricarboxylic acid (TCA) cycle or the ATP synthase inhibitor oligomycin A to inhibit oxidative phosphorylation (*Figure 8E*). Quantitative analysis of percentage of gliding motility showed either glucose or glutamine significantly increased gliding motility by bradyzoites (*Figure 8F*), indicating that either carbon source can be used to produce ATP for maintaining gliding motility. Either 2-DOG or oligomycin A blocked gliding motility by bradyzoites even in the presence of exogenous glucose or glutamine (*Figure 8F*), demonstrating that both oxidative phosphorylation and glycolysis are ATP sources for driving gliding motility by bradyzoites. Interestingly, oligomycin A blocked glucose-dependent gliding (*Figure 8F*), indicative of an essential role of mitochondrial electron transport chain in ATP production for supporting gliding motility of bradyzoites.

To further investigate the energy status of bradyzoites, we utilized reversed-phase high-performance liquid chromatography (RP-HPLC) to measure the adenosine triphosphate (ATP), adenosine diphosphate (ADP), and adenosine monophosphate (AMP) levels in bradyzoites treated with EC buffer containing both Ca$^{2+}$ (1.8 mM) and glucose (5.6 mM) for different times (*Figure 8—figure supplement 1A-C*). We observed that after the incubation in EC buffer for 1 hr, bradyzoites had significantly higher ATP, ADP, and AMP levels (*Figure 8G*), demonstrating enhanced ATP production during incubation. In contrast, ATP and ADP levels did not change significantly in tachyzoites despite the increase in AMP level after incubation with exogenous Ca$^{2+}$ and glucose (*Figure 8G*). The ATP/ADP ratio and energy charge have been widely used to evaluate cellular energy status, which controls the free-energy change for ATP hydrolysis for different cellular functions (*Maldonado and Lemasters, 2014*). Bradyzoites incubated with EC buffer for 1 hr displayed significantly increased ATP/ADP ratio and energy charge while no obvious changes were observed in tachyzoites (*Figure 8H and I*), indicating that bradyzoites rapidly recover their energy status following incubation with glucose. Collectively, exogenous Ca$^{2+}$ and glucose altogether activate bradyzoite gliding motility via restoration of ATP levels and Ca$^{2+}$ stores.

## Discussion

Calcium signaling plays important roles in the control of microneme secretion, gliding motility, and egress of apicomplexan parasites, and these pathways have been extensively characterized in the tachyzoite stage of *T. gondii* (*Lourido and Moreno, 2015*; *Hortua Triana et al., 2018*), although not widely explored in other motile life-cycle stages. Here, we compared the responses of *T. gondii* tachyzoites and bradyzoites to Ca$^{2+}$ ionophores and agonists that cause release of Ca$^{2+}$ from intracellular stores and found that Ca$^{2+}$ responses, microneme secretion, and egress by bradyzoites were all highly attenuated. Dampened Ca$^{2+}$ responses were evident in the responses of in vitro cysts differentiated under stress conditions, naturally occurring cysts formed in muscle cells, and tissue cysts purified from brains of chronically infected mice and tested ex vivo. Reduced responses were not simply a consequence of the cyst environment as similar dampened Ca$^{2+}$ signals and microneme secretion were observed in single, extracellular bradyzoites. Ratiometric Ca$^{2+}$ imaging revealed lower resting Ca$^{2+}$ levels and reduced ER and acidic-stored Ca$^{2+}$ in bradyzoites, which is likely a reflection of downregulation of Ca$^{2+}$-ATPases involved in maintaining these stores replenished. Tissue cysts are characterized by a thick wall comprising proteins and carbohydrates that may collectively impede signals and/or restrict egress mechanically. However, when cysts were digested by trypsin to release bradyzoites, they exhibited Ca$^{2+}$-dependent gliding motility that was enhanced by incubation in extracellular Ca$^{2+}$ in combination with glucose, demonstrating that they express a conserved mechanism for Ca$^{2+}$ mediated motility, albeit dampened by reduced stored Ca$^{2+}$ and diminished energy levels. The dampened Ca$^{2+}$ signaling responses of bradyzoites reflect adaptations that are well suited to the long-term intracellular lifestyle of these chronic stages. Significantly, bradyzoites also retain the potential to become motile once provided with sources of energy and Ca$^{2+}$, demonstrating remarkable physiological flexibility that favors transmission.

Egress is a crucial step in the lytic cycle of apicomplexan parasites, and this response requires the sequential steps of increase in cytoplasmic Ca$^{2+}$, secretion of micronemes, PV rupture, and activation of motility (*Frénal et al., 2017*; *Carruthers, 2019*). Our studies demonstrate that bradyzoites show minimal egress from in vitro-differentiated cysts in response to agonists that normally trigger this response in tachyzoites (i.e., Ca$^{2+}$ ionophores and zaprinast). We also demonstrate that bradyzoites are refractory to stimulation of microneme secretion using either an intracellular reporter monitoring the release of PLP1 based on the dispersion of FNR-mCherry from the cyst matrix or a MIC2-GLuc reporter detecting secretion from extracellular bradyzoites. To explore the basis for these differences, we utilized a dual fluorescent reporter GCaMP6f BAG1-mCherry to monitor changes of cytosolic Ca$^{2+}$ levels in bradyzoites. Calcium signaling was significantly dampened in bradyzoites as reflected in delayed Ca$^{2+}$ spikes and lower magnitude of cytosolic Ca$^{2+}$ increases in response to Ca$^{2+}$ agonists. Reduced Ca$^{2+}$ responses were also confirmed using bradyzoites naturally formed in C2C12 skeletal muscle cells and ex vivo cysts isolated from chronically infected mice, indicating that the dampened responses are not simply a consequence of alkaline pH stress during bradyzoites development in vitro. Additionally, we observed similar dampened responses from extracellular bradyzoites, indicating that decreased responses are not simply due to reduced permeability of intact cysts to agonists. To

confirm these results, we also utilized Fluo-8/AM to monitor intracellular $Ca^{2+}$ stores of bradyzoites and observed similar dampened responses. Finally, since $Ca^{2+}$-dependent fluorescence responses by GCaMP6f or Fluo-8 are only relative and subject to differences in protein or probe levels, we developed a ratiometric $Ca^{2+}$ reporter that contains GCaMP6f fused with self-cleavage tag P2A-linked mTagBFP2 under the control of the same promoter. Ratiometric measurements of the GCaMP6f signal compared to the $Ca^{2+}$ insensitive indicator mTagBFP2 determined that bradyzoites have lower resting $Ca^{2+}$ levels and quantitatively decreased $Ca^{2+}$ responses relative to tachyzoites in response to $Ca^{2+}$ agonists. Collectively, these findings conclusively show that bradyzoites have reduced $Ca^{2+}$ responses whether developed in vitro or in vivo and using a variety of independent methods to assess both $Ca^{2+}$ levels and physiological responses.

Based on the above findings, it seems likely that bradyzoites possess different mechanisms to control $Ca^{2+}$ homeostasis, including differences in expression of $Ca^{2+}$ channels and $Ca^{2+}$ pumps relative to tachyzoites. These differences would impact $Ca^{2+}$ storage pools, affecting cytosolic $Ca^{2+}$ and signaling. For example, our findings indicate that bradyzoites show reduced responses to ionomycin and thapsigargin, which release $Ca^{2+}$ primarily from the ER, and in response to $NH_4Cl$, which releases $Ca^{2+}$ from acidocalcisomes and likely other acidic stores (*Moreno and Zhong, 1996*; *Stasic et al., 2021*). Consistent with these dampened responses, bradyzoites showed significantly reduced expression of the $Ca^{2+}$-ATPases TgSERCA (*Nagamune et al., 2007*) and TgA1 (*Luo et al., 2004*), which are involved in transporting cytosolic $Ca^{2+}$ into the ER and acidocalcisome, respectively. They also showed reduced expression of TgA2, the $Ca^{2+}/H^+$ exchanger and the recently described TRPPL-2 (*Márquez-Nogueras et al., 2021*), which is a TRP channel key for cytosolic $Ca^{2+}$ influx through the plasma and ER membranes. The reduced expression of $Ca^{2+}$ channels that allow influx into the cytosol and reduced expression of $Ca^{2+}$ pumps that fill intracellular stores would result in a general reduction of stored $Ca^{2+}$. Additionally, it is possible that the reduced levels of $Ca^{2+}$ in bradyzoites reflect limitations on the availability of $Ca^{2+}$ from the host cell since prior studies have shown that tachyzoites acquire their intracellular $Ca^{2+}$ from this source (*Vella et al., 2021*). Reduced ER $Ca^{2+}$ could impact mitochondrial $Ca^{2+}$ since it has been shown in mammalian cells that $Ca^{2+}$ can be transferred directly (through membrane contact sites) from the ER to the mitochondria (*Cárdenas et al., 2010*; *Gherardi et al., 2020*), which is essential for oxidative phosphorylation and ATP production. One aspect that is not addressed by our studies is whether altered expression of $Ca^{2+}$ channels and pumps is responsible for reducing energy levels and hence driving quiescence, or whether the altered $Ca^{2+}$ pathways are a consequence of initial changes in energy production. Further studies will be needed to decipher the contribution of these various mechanism to altered $Ca^{2+}$ homeostasis and signaling in bradyzoites.

Bradyzoites are surrounded by a cyst wall that comprises an outer thin compact layer and an inner sponge-like layer that faces the cyst matrix (*Lemgruber et al., 2011*). The cyst wall is enriched in dense granule proteins (*Tu et al., 2019*), stage-specific glycoproteins such as CST1 (*Petri et al., 2001*; *Tomita et al., 2013*), and partially characterized carbohydrates (*Tomita et al., 2017*). This architecture may create a barrier to egress since bradyzoites were able to activate motility but not to efficiently emerge from intact cysts. We utilized trypsin to digest the cyst wall, mimicking the cyst rupture observed in chronically infected mice or following oral ingestion and exposure to pepsin (*Ferguson et al., 1989*; *Dubey, 1998*). Notably, proteolytic release did not result in immediate changes in $Ca^{2+}$ nor motility in the parasite, suggesting that cyst wall degradation does not trigger a process akin to egress in tachyzoites. Rather, when artificially released in this manner, a subset of bradyzoites spontaneously underwent gliding motility associated with $Ca^{2+}$ oscillations that were similar to those previously described for tachyzoites (*Lovett and Sibley, 2003*). When incubated with extracellular $Ca^{2+}$, the percentage of motile bradyzoites increased dramatically, suggesting that $Ca^{2+}$ entry stimulates motility, similar to tachyzoites (*Pace et al., 2014*; *Borges-Pereira et al., 2015*). Unlike a previous report showing that tachyzoites contain sufficient $Ca^{2+}$ stores and energy levels to be independent of external carbon sources during the first hour after liberation (*Lin et al., 2011*), we observed that bradyzoites require an external source of carbon to regain $Ca^{2+}$ stores and ATP levels. Similar to previous findings that *T. gondii* tachyzoites can support motility either from glucose through glycolysis or from glutamine that feeds into the TCA cycle (*Blume et al., 2009*; *MacRae et al., 2012*), we observed that either carbon source was capable of synergizing with $Ca^{2+}$ to restore bradyzoite motility, although inhibitor studies indicate that oxidative phosphorylation is required to restore optimal energy levels. Consistent with this prediction, we observed that bradyzoites have intrinsically low ATP/ADP ratios but that they

recovered substantially when incubated extracellularly for 1 hr in $Ca^{2+}$ and glucose. Collectively, these findings indicate that bradyzoites are characterized by both low $Ca^{2+}$ stores and low ATP levels, but that they respond to changes in the extracellular environment to restore both energy levels and $Ca^{2+}$ signaling systems needed for motility. During natural egress, it is also possible that bradyzoites rely on endogenous energy stores such as amylopectin, especially since $Ca^{2+}$ levels and CDPK2 have been shown to influence the storage and utilization of this carbohydrate (*Uboldi et al., 2015*). Stimulation of $Ca^{2+}$ signaling is also important in breaking dormancy (*Pang et al., 2007*) and pollen germination in plants (*Steinhorst and Kudla, 2013*), and initiation of the cell cycle in animal cells (*Humeau et al., 2018*), demonstrating the important role played by $Ca^{2+}$ signaling in reactivation.

Reduced $Ca^{2+}$ storage, dampened $Ca^{2+}$ signaling, and a lower energy state may reflect the long-term sessile nature of the intracellular cyst, which prolong chronic infection. The mechanisms inducing cyst wall turnover in vivo are unclear, although host cell macrophages may contribute to this process as they secrete chitinase that can lyse cysts in vitro (*Nance et al., 2012*). Additionally, cyst wall turnover may be controlled by release of parasite hydrolases as suggested by the presence of GRA56, which is predicted to belong to the melibiase family of polysaccharide degrading enzymes, on the cyst wall (*Nadipuram et al., 2020*). Our in vitro studies suggest that once the cyst wall is ruptured bradyzoites respond to higher levels of $Ca^{2+}$ and glucose in the extracellular environment to regain motility needed for subsequent cell invasion. Emergence of bradyzoites from tissue cysts that rupture in muscle or brain, or in tissue following oral ingestion, is likely to provide an environment to recharge bradyzoites. Consistent with this idea, previous in vitro studies have shown that similar motile bradyzoites released from ruptured cysts have the ability to reinvade new host cells, establishing new cysts without an intermediate growth stage as tachyzoites (*Dzierszinski et al., 2004*). Hence, the rapid metabolic recovery of otherwise quiescent bradyzoites may be important for the maintenance of chronic infection within a single host and to assure robust cellular invasion upon transmission to the next host.

## Materials and methods

**Key resources table**

| Reagent type (species) or resource | Designation | Source or reference | Identifiers | Additional information |
|---|---|---|---|---|
| Recombinant DNA reagent | pBAG1:mCherry, SAG1:CAT, TUB1:GCaM6f (pNJ-26) | This paper | | Generation of BAG1-mCherry GCaMP6f reporter |
| Recombinant DNA reagent | pSAG1:CAS9-EGFP, U6:sgUPRT | Addgene | Addgene_54467 | Template for construction of pSAG1:CAS9-GFP, U6:sgDHFR 3'UTR |
| Recombinant DNA reagent | pSAG1:CAS9-EGFP, U6:sgDHFR 3'UTR | This paper | | Generation of ratiometric reporter |
| Recombinant DNA reagent | p2A-mTagBFP2, DHFR-TS:HXGPRT | This paper | | Generation of ratiometric reporter |
| Recombinant DNA reagent | pBAG1:EGFP, DHFFR-TS::HXGPRT | This paper | | Generation of BAG1-EGFP reporter |
| Recombinant DNA reagent | pBAG1:mCherry, DHFFR-TS::HXGPRT | This paper | | Generation of BAG1-mCherry reporter |
| Recombinant DNA reagent | pMIC2:GLuc-myc, DHFR-TS | *Brown et al., 2016* | | Generation of MIC2 secretion reporter |
| Recombinant DNA reagent | pTUB1:FNR-mCherry, CAT | Other | | Vernon Carruthers Lab in University of Michigan |
| Recombinant DNA reagent | pTUB1:YFP-mAID-3HA, DHFR-TS:HXGPRT | Other, *Brown et al., 2017* | | Template for construction of plasmids in this paper |
| Transfected construct (*Toxoplasma gondii*) | ME49 Δhxgprt::FLUC | Other, *Tobin et al., 2012* | | Parental stain for generation of reporters in this paper |

*Continued on next page*

*Continued*

| Reagent type (species) or resource | Designation | Source or reference | Identifiers | Additional information |
|---|---|---|---|---|
| Transfected construct (*T. gondii*) | BAG1-mCherry GCaMP6f | This paper | | Genotypes are indicated as ME49 Δ*hxgprt::TUB1:FLUC; BAG1:mCherry, SAG1:CAT, TUB1:GCaMP6f* |
| Transfected construct (*T. gondii*) | BAG1-mCherry | This paper | | Genotypes are indicated as ME49 Δ*hxgprt::TUB1:FLUC; BAG1:mCherry, DHFR-TS:HXGPRT* |
| Transfected construct (*T. gondii*) | BAG1-EGFP | This paper | | Genotypes are indicated as ME49 Δ*hxgprt::TUB1:FLUC; BAG1:EGFP, DHFR-TS:HXGPRT* |
| Transfected construct (*T. gondii*) | BAG1-mCherry MIC2-GLuc | This paper | | Genotypes are indicated as ME49 Δ*hxgprt::TUB1:FLUC; BAG1:mCherry, DHFR-TS:HXGPRT; MIC2:MIC2-GLuc, DHFR-TS* |
| Transfected construct (*T. gondii*) | BAG1-EGFP FNR-mCherry | This paper | | Genotypes are indicated as ME49 Δ*hxgprt::TUB1:FLUC; BAG1:EGFP, DHFR-TS:HXGPRT; SAG1:CAT, TUB1:FNR-mCherry* |
| Transfected construct (*T. gondii*) | BAG1-mCherry GCaMP6f-P2A-mTagBFP2 | This paper | | Genotypes are indicated as ME49 Δ*hxgprt::TUB1:FLUC; BAG1:mCherry, SAG1:CAT, TUB1:GCaMP6f-P2A-mTagBFP2, DHFR-TS:HXGPRT* |

## Cell culture

*T. gondii* tachyzoites were passaged in confluent monolayers of HFFs obtained from the Boothroyd Laboratory at Stanford University. The ME49 Δ*hxgprt::Fluc* type II strain of *T. gondii* (*Tobin et al., 2012*) was used as a parental strain for genetic modification. Tachyzoites were cultured in Dulbecco's modified Eagle's medium (DMEM; Life Technologies) pH 7.4, supplemented with 10% fetal bovine serum (FBS), penicillin, and streptomycin (Life Technologies) at 37°C in 5% $CO_2$. Parasite and host cell lines were confirmed to be negative for mycoplasma using an e-Myco plus kit (Intron Biotechnology). For in vitro induction of bradyzoites, parasites were cultured in alkaline medium in ambient $CO_2$ as described previously (*Wang et al., 2015*). In brief, infected HFF monolayers were switched to RPMI 1640 medium (MP Biomedicals) buffered to pH 8.2 with HEPES and supplemented with 5% FBS and cultured at 37°C in ambient $CO_2$, during which time the alkaline medium was changed every 2 days. For spontaneous induction of bradyzoites, C2C12 muscle myoblast cells (ATCC CRL-1772) were maintained in DMEM supplemented with 20% FBS. C2C12 myoblast differentiation and myotube formation were induced in DMEM containing 2% horse serum (Biochrom) by cultivation at 37°C in 5% $CO_2$ for 5 days. Tachyzoites were inoculated into the differentiated muscle cells and cultured for another 7 days to induce bradyzoite formation, during which time the induction medium was changed every 2 days. For harvesting bradyzoites, infected monolayers were scraped into intracellular (IC) buffer (142 mM KCl, 5 mM NaCl, 1 mM $MgCl_2$, 5.6 mM D-glucose, 2 mM EGTA, 25 mM HEPES, pH 7.4) and released from cells by serially passing through 18G, 20G, and 25G needles, followed by centrifugation (150 × *g*, 4°C) for 10 min. The pellet containing cysts was resuspended in IC buffer. Bradyzoites were liberated from cysts by digestion with 0.25 mg/ml trypsin at room temperature for 5 min, followed by centrifugation (150 × *g*, 4°C) for 10 min. The supernatant containing liberated bradyzoites was further centrifuged (400 × *g*, 4°C) for 10 min. The pellet containing purified bradyzoites was resuspended in extracellular (EC) buffer (5 mM KCl, 142 mM NaCl, 1 mM $MgCl_2$, 5.6 mM D-glucose, 25 mM HEPES, pH 7.4) with (1.8 mM $Ca^{2+}$) or without $CaCl_2$, as indicated for different assays and in the legends.

## Purification of bradyzoites by magnetic beads

Tachyzoites were induced to form bradyzoites at a multiplicity of infection (MOI) of 0.5 by culture at pH 8.2 in RPMI 1640 medium under ambient air (low $CO_2$) for 7 days followed by scraping into PBS containing 0.1% bovine serum albumin (BSA). Cysts were released from host cells by repeated passage through a 23G needle and collected by centrifugation at 150 × *g* for 10 min. To purify large-scale bradyzoites used for western blotting, pelleted cysts were resuspended in 1 ml PBS containing 10 µl biotinylated *Dolichos biflorus* agglutinin (DBA) (Vector Laboratories) and 100 µl Pierce Streptavidin Magnetic Beads (Thermo Fisher) and incubated for 1 hr at 4 °C. The beads and absorbed cysts were collected using a magnetic stand and resuspended in 1 ml PBS containing 0.25 mg/ml trypsin

and incubated for 10 min at 16°C. The supernatant containing released bradyzoites was separated from the beads and retained. To remove any residual tachyzoites in the supernatant, 5 µl of mAb DG52 pre-coupled to 100 µl Dynabeads Protein G (Thermo Fisher) was added to the supernatant and incubated for 1 hr at 4 °C. The supernatant was separated from the beads, bradyzoites centrifuged at 600 × $g$, 4°C for 10 min and kept for different assays. To purify bradyzoites used for investigation of recovery of microneme secretion and high-performance liquid chromatography UV (HPLC-UV) analysis of ATP, ADP, and AMP levels in bradyzoites, PBS was replaced with EC buffer without $Ca^{2+}$ or glucose during the purification by magnetic beads.

## Reagents and antibodies

A23187, zaprinast, ionomycin, thapsigargin, $NH_4Cl$, fluorescein isothiocyanate-conjugated DBA, and BAPTA-AM were obtained from Sigma. Fluo-8 AM was obtained from Abcam. SYTOX Red Dead Cell Stain was obtained from Thermo Fisher. The compounds 3-MB-PP1 (*Lourido et al., 2010*) and compound 1 (*Brown et al., 2016*) were obtained as described previously. Trypsin and L-glutamine were purchased from MP Biomedicals. Adenosine 5'-triphosphate (ATP) disodium salt, adenosine 5'-diphosphate (ADP) sodium salt, adenosine 5'-monophosphate (AMP) disodium salt, oligomycin A, and 2-DOG were purchased from Sigma. Primary antibodies include mouse mAb DG52 anti-SAG1 (provided by John Boothroyd), mouse mAb 6D10 anti-MIC2 (*Carruthers et al., 1999a*), rabbit anti-GRA7 (*Alaganan et al., 2014*), mouse mAb 8.25.8 anti-BAG1 (obtained from Louis Weiss), rabbit anti-BAG1 (obtained from Louis Weiss), rabbit anti-M2AP (obtained from Vernon B. Carruthers), mouse anti-c-myc (mAb 9E10, BioLegend), mouse anti-acetylated Tubulin (mAb 6-11B-1, Sigma), rat anti-mCherry (mAb 16D7, Life Technologies), rabbit-anti SRS9 (obtained from John Boothroyd), rabbit anti-tRFP (Axxora), and mouse anti-6XHis (mAbHIS.H8, Life Technologies). Secondary antibodies for IFAs include goat anti-mouse IgG conjugated to Alexa Fluor-488, goat anti-rabbit IgG conjugated to Alexa Fluor-488, anti-mouse IgG conjugated to Alexa Fluor-568, goat anti-rat IgG conjugated to Alexa Fluor-568, and goat anti-mouse IgG conjugated to Alexa Fluor-594 (Life Technologies). For western blotting, secondary antibodies consisted of goat anti-mouse IgG, goat anti-rabbit IgG, or goat anti-rat IgG conjugated to LI-COR C800 or C680 IR-dyes and detected with an Odyssey Infrared Imaging System (LI-COR Biotechnology).

## Generation of stable transgenic parasite lines

### Dual $Ca^{2+}$ and bradyzoite reporter strain: BAG1-mCherry GCaMP6f

A dual reporter designed to detect bradyzoite conversion and $Ca^{2+}$ fluctuations was generated in the ME49 *Δhxgprt::Fluc* strain (*Tobin et al., 2012*). We generated a plasmid named pNJ-26 that contains mCherry driven by the BAG1 promoter, the genetically encoded $Ca^{2+}$ indicator GCaMP6f under the control of Tubulin1 promoter, and the selection marker cassette SAG1 promoter driving CAT. ME49 *Δhxgprt::Fluc* tachyzoites were transfected with 20 µg of the pNJ-26 plasmid and selected with 20 µM chloramphenicol. Clones containing randomly integrated transgenes were confirmed by diagnostic PCR and by IFA staining. Primers are listed in *Supplementary file 1*.

### Bradyzoite reporter strain: BAG1-EGFP and BAG1-mCherry

The BAG1 promoter and the mCherry open reading frame (ORF) were independently PCR-amplified from pNJ-26 and the EGFP ORF was amplified from pSAG1:CAS9-U6:sgUPRT, respectively. The BAG1 promoter fragment and EGFP ORF or mCherry (ORF) were cloned by NEBuilder HiFi DNA Assembly Cloning Kit (NEB, E5520S) into the vector backbone that was produced by double enzymatic digestion of pTUB1:YFP-mAID-3HA, DHFR-TS:HXGPRT using KpnI and NdeI. ME49 *Δhxgprt::Fluc* tachyzoites were transfected with 20 µg pBAG1:EGFP, DHFFR-TS:HXGPRT or pBAG1:mCherry, DHFFR-TS:HXGPRT and selected with mycophenolic acid (MPA) (25 µg/ml) and 6-xanthine (6Xa) (50 µg/ml). Single-cell clones containing randomly integrated transgenes were confirmed by diagnostic PCR and by IFA staining. Primers are listed in *Supplementary file 1*.

### MIC2 secretion reporter BAG1-mCherry MIC2-GLuc

The bradyzoite reporter line BAG1-mCherry was transfected with 20 µg of the previously described pMIC2:GLuc-myc, DHFR-TS plasmid (*Brown et al., 2016*), and selected with 3 µM pyrimethamine

(PYR). Single-cell clones containing randomly integrated transgenes were confirmed by diagnostic PCR and by IFA staining.

## FNR-mCherry leakage reporter BAG1-EGFP FNR-mCherry

The bradyzoite reporter line BAG1-EGFP was transfected with 20 µg pTUB1:FNR-mCherry, CAT (provided by the Carruthers lab), and selected with 20 µM chloramphenicol. Single-cell clones containing randomly integrated transgenes were confirmed by diagnostic PCR and IFA staining.

## Ratiometric reporter BAG1-mCherry GCaMP6f-P2A-mTagBFP2

The ratiometric reporter strain was generated using targeted insertion with CRISPR/Cas9 using previously described methods (*Shen et al., 2017*) to add the BFP downstream of the GCaMP6f protein in the strain BAG1-mCherry GCaMP6f. In brief, a single-guide RNA (sgRNA) targeting the DHFR 3′UTR following the GCaMP6f coding sequence was generated in the plasmid pSAG1:CAS9-U6:sgUPRT (*Shen et al., 2014*). The P2A-mTagBFP2 tagging plasmid was constructed by cloning a synthetic sequence containing a slit peptide (P2A) together with the blue fluorescent reporter mTagBFP2 (P2A-mTagBFP2) into the pTUB1:YFP-mAID-3HA, DHFR-TS:HXGPRT backbone by NEBuilder HiFi DNA Assembly Cloning Kit (NEB, E5520S) after double enzymatic digestion of KpnI and NdeI. Following this step, the SAG1 3′UTR was amplified from pNJ-26 and cloned into the tagging plasmid to replace DHFR 3′UTR by Gibson assembly (NEB, E5520S). BAG1-mCherry GCaMP6f reporter tachyzoites were co-transfected with 10 µg of pSAG1::CAS9-U6::sgDHFR 3′UTR and 2 µg of PCR-amplified P2A-mTagBFP2-HXGPRT flanked with 40 bp homology regions, as described previously (*Long et al., 2017*). Stable transfectants were selected with 25 µg/ml MPA and 50 µg/ml 6Xa. Single-cell clones containing targeted integrated transgenes were confirmed by diagnostic PCR and IFA staining. Primers are shown in *Supplementary file 1*.

## Time-lapse imaging of fluorescent reporter strains

For time-lapse microscopy, extracellular parasites were added to glass-bottom culture dishes (MatTek) or intracellular parasites were grown in host cells attached glass-bottom culture dishes. Alternating phase and fluorescence images (at different intervals specified in the legends) were collected on a Zeiss AxioObserver Z1 (Carl Zeiss, Inc) equipped with an ORCA-ER digital camera (Hamamatsu Photonics) and a ×20 EC Plan-Neofluar objective (N.A. 0.50), 37°C heating unit, and LED illumination for blue, green, red, and far-red wavelengths. Spinning disc images were acquired with a ×100 oil Plan-Apochromat (N.A. 1.46) objective using illumination from 488 nm and 561 nm solid-state lasers (Zeiss) and Evolve 512 Delta EMCCD cameras (Photometrics) attached to the same Zeiss AxioObserver Z1 microscope. Images were acquired and analyzed using Zen software 2.6 blue edition (Zeiss). Fluorescent intensity changes ($F/F_0$) vs. time were plotted with GraphPad Prism version 6 (GraphPad Software, Inc).

## Indirect IFA

Parasites grown in HFF monolayers on glass coverslips were fixed in 4% (v/v) formaldehyde in PBS for 10 min, permeabilized by 0.25% (v/v) Triton X-100 in PBS for 20 min, and blocked in 3% BSA in PBS. Monolayers were incubated with different primary antibodies and visualized with secondary antibodies conjugated to Alexa Fluor. Coverslips were sealed onto slides using ProLong Gold Antifade containing DAPI (Thermo Fisher Scientific). Images were captured using a ×63 oil Plan-Apochromat lens (N.A. 1.4) on an Axioskop2 MOT Plus Wide Field Fluorescence Microscope (Carl Zeiss, Inc). Scale bars and linear adjustments were made to images using Axiovision LE64 software (Carl Zeiss, Inc).

## Western blotting

Samples were prepared in 5× Laemmli buffer containing 100 mM dithiothreitol, boiled for 5 min, separated on polyacrylamide gels by SDS-PAGE, and transferred to nitrocellulose membrane. Membranes were blocked with 5% nonfat milk and probed with primary antibodies diluted in blocking buffer. Membranes were washed with PBS + 0.1% Tween 20, then incubated with goat IR dye-conjugated secondary antibodies (LI-COR Biosciences) in blocking buffer. Membranes were washed several times before scanning on a LI-COR Odyssey imaging system (LI-COR Biosciences).

## Fluo-8 AM Ca²⁺ monitoring

Freshly harvested parasites were loaded with 500 nM Fluo-8 AM for 10 min at room temperature, followed by centrifugation at 400 × $g$ for 5 min and washing in EC buffer without Ca²⁺. Parasites were resuspended in EC buffer without Ca²⁺ and added directly to glass-bottom culture dishes. After addition of agonists, time-lapse images were recorded and analyzed as described above.

## Egress assay

Infected cells were treated with 2 µM A23187 or 500 µM zaprinast for 15 min at 37°C. Following incubation, samples were stained by IFA using antibodies against SAG1 (mouse), GRA7 (rabbit), FITC-conjugated DBA or BAG1 (rabbit), and followed by secondary antibodies conjugated to Alexa Fluor. Samples were examined by fluorescence microscopy, and the percentages of egressed or released parasites per vacuole or cyst was determined at least for 20 vacuoles or cysts per experiment. The maximum egress distance of parasites from vacuole or cysts was measured from scanned tiff images in ImageJ.

## Flow cytometry

ME49 BAG1-mCherry MIC2-GLuc reporter bradyzoites were induced for 7 days at pH 8.2, harvested in IC buffer as described above, and passed through a 3 µm polycarbonate membrane filter. ME49 Δ*hxgprt::Fluc* tachyzoites, cultured and harvested as indicated above, were used for gating. Approximately 1 × 10⁶ parasites from each sample (ME49 BAG1-mCherry MIC2-GLuc reporter tachyzoites and ME49 BAG1-mCherry MIC2-GLuc reporter bradyzoites) were sorted on Sony SH800S Cell Sorter directly into 500 µl IC buffer followed by centrifugation. Flow cytometry data were processed using FlowJo version 10 (FlowJo, LLC).

## Collection of excretory-secretory antigens (ESA) and *Gaussia* luciferase assay

FACS-sorted MIC2-GLuc reporter tachyzoites and bradyzoites, or bradyzoites purified by magnetic beads, were suspended with EC buffer and incubated with different agonists at 37°C for 10 min. ESA was collected by centrifugation and mixed with Pierce *Gaussia* Luciferase Glow Assay Kit reagent (Thermo Scientific), and luminescence was detected using a Cytation 3 Cell Imaging Multimode Imager (BioTek Instruments, Inc). Buffer control values were subtracted from their corresponding sample values to correct for background.

## Real-time PCR

RNA was extracted from ME49 Δ*hxgprt::Fluc* tachyzoites and bradyzoites induced for 7 days at pH 8.2 using RNeasy Mini Kit (QIAGEN) combined with QIAshredder (QIAGEN) followed by DNA Removal using DNA-free DNA Removal Kit (Thermo Fisher) and subsequent reverse transcription using High-Capacity cDNA Reverse Transcription Kit (Thermo Fisher). Quantitative real-time PCR was performed on Applied Biosystems QuantStudio 3 Real-Time PCR System (Thermo Fisher) using SYBR Green JumpStart Taq ReadyMix (Sigma) with primers shown in *Supplementary file 1*. Mean fold changes from two independent experiments were calculated from ΔΔ Ct values using actin1 transcript as housekeeping gene, as described previously (*Livak and Schmittgen, 2001*).

## Gliding trail assay

Coverslips were precoated by incubation in 50% FBS diluted in PBS for 1 hr at 37°C followed by rinsing in PBS. Freshly harvested tachyzoites or bradyzoites were resuspended in EC buffer, treated with DMSO (0.1%, v/v), or inhibitors (in 0.1% DMSO, v/v), and then added to precoated glass coverslips and incubated at 37°C for 15 min. Coverslips were fixed in 2.5% formalin in PBS for 10 min, and the surface proteins were detected by IFA as described above using anti-SAG1 and anti-SRS9 antibodies as stage-specific markers for tachyzoites and bradyzoites, respectively. Gliding trails were captured by IFA microscopy as described above, and the frequency of trails was measured from tiff images using ImageJ.

## Gliding motility assay based on time-lapse video microscopy

BAG1-mCherry parasites were induced to form bradyzoites by culture at pH 8.2 in RPMI 1640 medium under ambient air (low $CO_2$) for 7 days followed by scraping into EC buffer without Ca²⁺ or

glucose and repeated passage through a 23G needle. Intact but extracellular cysts were pellet by centrifugation at 150 × g for 10 min and resuspended in EC buffer without $Ca^{2+}$ or glucose and incubated for ~2 hr at 4°C. MatTek 25 mm glass-bottom dishes (coverslip dishes) were precoated with 2 ml 50% FBS at 4°C overnight and rinsed twice using PBS prior to use. Purified cysts were added to the precoated coverslip dishes in EC buffer without $Ca^{2+}$ or glucose but containing 0.25 mg/ml trypsin and incubated for 10 min at 16°C. The medium was removed and 2 ml EC buffer ± 1.8 mM $Ca^{2+}$ and/or ±5.6 mM glucose was added and incubated for 10 min or 1 hr at 16°C. Prior to imaging, the coverslip dishes were heated to 37°C using a Heating Unit XL S (Zeiss) attached to the Zeiss AxioObserver Z1 (Carl Zeiss, Inc). Images were collected under bright-field illumination using a ×40 C-Apochromat water immersion objective (N.A. 1.20) and ORCA-ER digital camera (Hamamatsu Photonics) at 1 s intervals for 5 min per field. The percentage of BAG1-mCherry-positive bradyzoites displaying different types of gliding motility was calculated from six movies per sample. Images were imported into NIH ImageJ with a Cell Counter plug-in for quantification of the types of motility based on visual inspection.

## HPLC-UV analysis of ATP, ADP, and AMP levels in bradyzoites

Bradyzoites were induced for 7 days at alkaline pH and purified by magnetic beads as described above, followed by resuspension in 1 ml EC buffer containing 1.8 mM $Ca^{2+}$ and 5.6 mM glucose for 10 min or 1 hr at 16°C. As control, tachyzoites were harvested and incubated in EC buffer without $Ca^{2+}$ or glucose for 2 hr at 4°C, followed by treatment with EC buffer containing 1.8 mM $Ca^{2+}$ and 5.6 mM glucose for 10 min or 1 hr 16°C. Following incubation, parasites were pelleted at 600 × g, 4°C for 10 min, and stored at –80°C until analysis.

A previously described method for extraction of ATP, ADP, and AMP (Menegollo et al., 2019) was adapted for use here. In brief, 95 µl of extraction buffer (0.3 M perchloric acid [$HClO_4$], 1 mM ethylenediaminetetraacetic acid disodium salt [$Na_2EDTA$], pH 8.0) was used to resuspend cell pellets and incubated for 5 min at room temperature. Extraction was stopped by addition of 17 µl of neutralization buffer (2 M potassium hydroxide) to the samples followed by mixing. Samples were centrifuged at 14,000 × g for 10 min at 4°C, and the supernatant was transferred to a new tube for HPLC analysis. Analysis was performed using an HPLC system consisting of a SPD-20A UV/VIS detector (Shimadzu) equipped with SIL-20A autosampler (Shimadzu), with a Luna Omega Polar C18 column (4.6 mm internal diameter × 150 mm length, 3 µm particle size, 100 Å pore size) and LC-20AD pump (Shimadzu). The protocol was setup as isocratic separation using a mobile phase containing 0.1 M ammonium dihydrogen phosphate ($NH_4H_2PO_4$, Sigma), pH 6.0, containing 1% methanol with a flow rate of 0.8 ml/min. Injection volume was 30 µl, and peak detection was monitored at 254 nm. A series of standards containing ATP, ADP, and AMP with different concentrations were used to establish retention times and standard calibration curves by calculating peak area. Samples from two independent biological replicates were analyzed using three technical replicates. The retention time and peak areas were used to calculate the corresponding concentration of each nucleotide from each sample according to the standard curve.

## Mouse infections and ex vivo cyst collection

Mice were housed in an Association for Assessment and Accreditation of Laboratory Animal Care International-approved facility at Washington University School of Medicine. All animal studies were conducted in accordance with the U.S. Public Health Service Policy on Humane Care and Use of Laboratory Animals, and protocols were approved by the Institutional Animal Care and Use Committee at the School of Medicine, Washington University in St. Louis.

Eight-week-old female CD-1 mice (Charles River) were infected with 200 ME49 BAG1-mCherry GCaMP6f tachyzoites by intraperitoneal injection. After 30 days of infection, animals were sacrificed, the brain removed and homogenized, and the number of brain cyst was determined by DBA staining and microscopy as previously described (Wang et al., 2015). Eight-week-old female CD-1 mice (Charles River) were infected with five cysts from the brain homogenate by oral gavage. Following a 30-day period, these mice were euthanized, and brain homogenate was collected and added to glass-bottom dishes for live imaging of tissue cysts.

## Statistical analyses

Statistical analyses were performed in Prism (GraphPad). Data that passed normally distribution were analyzed by one-way ANOVA or Student's $t$-tests, while data that were not normally distributed, or contain too few samples to validate the distribution, were analyzed by Mann–Whitney or Kruskal–Wallis nonparametric tests. $*p<0.05$, $**p<0.01$, $***p<0.001$.

## Acknowledgements

We thank Jennifer Powers Carson for technical help with the HPLC analysis, which was performed in the Washington University Core Laboratory for Clinical Studies. We thank Vern Carruthers for providing plasmids and antibodies, Louis Weiss and John Boothroyd for providing antibodies, members of the Sibley lab for helpful advice, Alex Rosenburg for early efforts to develop the C2C12 muscle cell system and advice on how to implement it, Wandy Beatty, Microbiology Imaging Facility, for technical assistance with microscopy, and Jenn Barks for tissue culture support. This work was supported in part by a grant from the NIH (AI#034036).

## Additional information

### Funding

| Funder | Grant reference number | Author |
|---|---|---|
| National Institutes of Health | AI034036 | L David Sibley |
| National Institutes of Health | AI128356 | Silvia NJ Moreno |
| National Institutes of Health | AI143857 | L David Sibley |

The funders had no role in study design, data collection and interpretation, or the decision to submit the work for publication.

### Author contributions

Yong Fu, Conceptualization, Data curation, Formal analysis, Investigation, Methodology, Visualization, Writing - original draft, Writing – review and editing; Kevin M Brown, Nathaniel G Jones, Methodology, Resources, Writing – review and editing; Silvia NJ Moreno, Conceptualization, Writing – review and editing; L David Sibley, Conceptualization, Funding acquisition, Project administration, Supervision, Writing - original draft, Writing – review and editing

### Author ORCIDs

Nathaniel G Jones ⓘ http://orcid.org/0000-0001-7328-4487
Silvia NJ Moreno ⓘ http://orcid.org/0000-0002-2041-6295
L David Sibley ⓘ http://orcid.org/0000-0001-7110-0285

### Ethics

Mice were housed in an Association for Assessment and Accreditation of Laboratory Animal Care International-approved facility at Washington University School of Medicine. All animal studies were conducted in accordance with the U.S. Public Health Service Policy on Humane Care and Use of Laboratory Animals, and protocols were approved by the Institutional Animal Care and Use Committee at the School of Medicine, Washington University in St. Louis.

### Decision letter and Author response

Decision letter https://doi.org/10.7554/eLife.73011.sa1
Author response https://doi.org/10.7554/eLife.73011.sa2

## Additional files

### Supplementary files
• Supplementary file 1. Primers used in this study.

• Transparent reporting form

### Data availability
All of the data generated and analysed are included in the manuscript and supporting files including the meta data files.

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
