## [Editor Report]

This study shows that calcium signaling is strongly suppressed in the intracellular cyst-forming bradyzoite stages of *Toxoplasma gondii*. However, calcium signaling in these stages is rapidly restored following parasite egress and exposure to extracellular calcium and energy sources. The ability of *Toxoplasma* bradyzoites to rapidly switch between quiescence and a metabolically active state is likely to be essential for maintaining long-lived chronic infections as well as successful transmission.

---

## [Decision Letter]

**Decision letter after peer review:**

Thank you for submitting your article "Toxoplasma bradyzoites exhibit physiological plasticity of calcium and energy stores controlling motility and egress" for consideration by *eLife*. Your article has been reviewed by 3 peer reviewers, including Malcolm J McConville as the Reviewing Editor and Reviewer #1, and the evaluation has been overseen by a Reviewing Editor and Dominique Soldati-Favre as the Senior Editor. The following individual involved in review of your submission has agreed to reveal their identity: Christopher J Tonkin (Reviewer #2).

The three reviewers noted the novelty of these findings and the insights they provide into the biology of *T. gondii* bradyzoite stages. A number of issues were identified which may require additional experimental support and/or explicit discussion in the text.

Essential revisions:

1. Provide further validation of the Mic2-Gluc assay for measuring microneme secretion in bradyzoites – in particular it would be important to show that bradyzoites express sufficient M2AP to allow use of Mic2-Gluc as a read-out for secretion or confirm using an alternative assay.

2. Provide indication of variability in measurements of calcium levels where currently only the mean of just three cells is provided (or repeat with increased number of cells). This is important in Figure 5 where basal calcium levels are being measured.

3. Ensure that several points regarding methodology raised by the reviewers are directly addressed in the text, including the use of different time points when measuring ATP levels and gliding motility in extracellular bradyzoites, and discussion of possible effect of bradyzoite isolation protocol on calcium signalling.

4. Include discussion on the role of calcium signalling in mobilization of amylopectin carbohydrate stores and activation of bradyzoites in vivo

*Reviewer #1 (Recommendations for the authors):*

It remains unclear whether repression of calcium signalling in bradyzoites directs these stages into metabolic quiescence, or conversely, that entry into metabolic quiescence decreases the cellular energy balance with concomitant decrease in calcium uptake, depletion of intracellular stores and signalling. Can the authors draw out arguments for one side or the other?

As noted above, incubation of artificially released bradyzoites in calcium plus or minus a carbon source leads to increased motility and ATP levels. However, it is likely that activation of bradyzoites will occur earlier in vivo (e.g. prior to, or during the process of egress). During natural egress, bradyzoites will be more dependent on intracellular carbohydrate stores rather than exogenous sugars. Mobilization of internal stores is suggested by the finding that calcium alone is sufficient to activate motility in extracellular bradyzoites. Discussion of this possibility, including reference to previous work showing that calcium dependent kinases regulate amylopectin degradation is warranted.

Figure 8F. Can the authors provide information on the basal levels of ATP/ADP/AMP in tachyzoites suspended in the same EC buffer 10min/1hr, to allow comparison with bradyzoite data?

*Reviewer #2 (Recommendations for the authors):*

The introduction uses language that is a bit too definitive in places. I suggest to tone down.

– I would not say that ER is the known storage site of ca^2+^, I'd say the evidence suggests it is likely, but it is far from being definitive.

– Line 346- I would not say 1hr is a rapid change, especially in context of persistence in the brain or transmission (see below)

There are some spelling/grammatical mistakes; for example, Line 80: Accumulation of Ca^2+^ in the ER cannot lead to Mn secretion – this must be a mistake.

I think the discussion is too long and goes over a lot of what is already presented in Results section. I recommend reducing this but instead discuss more about how these results impact our understanding of transmission and persistence of a latent infection. For example, how do the lower levels of ca^2+^, motility and microneme secretion impact cyst lysis and re-invasion? Are these parameters enough or is there a transition period where bradyzoites become more motile to? Likewise, for transmission, it seems unlikely that an extracellular bradyzoite could hang around in the gut for 1 hr, to bolster energy and ca^2+^ needed to actively invade cells. Does this mean that high ca^2+^ and ATP (similar levels to tachyzoites) are not needed for transmission or do the authors think there is other possibilities? These considerations of the physiological relevance are important to discuss.

---

## [Author Response]

Essential revisions:1. Provide further validation of the Mic2-Gluc assay for measuring microneme secretion in bradyzoites – in particular it would be important to show that bradyzoites express sufficient M2AP to allow use of Mic2-Gluc as a read-out for secretion or confirm using an alternative assay.

We performed western blotting (Figure 2C) and IFA (Figure 2 supplement 1A) to confirm that bradyzoites express MIC2-Gluc and M2AP, albeit at lower levels compared with tachyzoites. Moreover, MIC2-GLuc and M2AP were properly co-localized to the apical end in bradyzoites, ruling out the possibility of mis-localization of the MIC2-GLuc reporter. Based on these results, we believe that MIC2-GLuc provides a reliable read-out for microneme secretion in in vitro differentiated bradyzoites. Additionally, the conclusion that MIC secretion is dampened in bradyzoites is also supported by the studies using the FNR-Cherry reporter in Figure 2E,F,G.

2. Provide indication of variability in measurements of calcium levels where currently only the mean of just three cells is provided (or repeat with increased number of cells). This is important in Figure 5 where basal calcium levels are being measured.

We have quantified more cells in all figures related to fluorescence measurements. For measurements of single parasites in Figure 5B, 5D, 5E, 6F, 8A, 8B and Figure 7 supplement 1A, we have now quantified 10 parasites for each condition and plotted the data as means ±S.D. For in vitro induced cysts or ex vivo cysts in Figure 2G, 3D, 3E, 4C,4G, 6E, 7B and Figure 4 supplement 1A, we measured 5 cysts or vacuoles per condition. Because these samples contain many parasites within each vacuole or cyst, they represent a greater sample size. The data are also plotted a means ±S.D.

3. Ensure that several points regarding methodology raised by the reviewers are directly addressed in the text, including the use of different time points when measuring ATP levels and gliding motility in extracellular bradyzoites, and discussion of possible effect of bradyzoite isolation protocol on calcium signalling.

We have conducted several new experiments to more closely match the experimental conditions to: (1) rule out that the differences are due to different time points or treatments (Figures 8G, 8H, 8I), (2) define the extent of bradyzoite differentiation (Figure 3 supplement 1A, 1B), (3) demonstrate the treatment with trypsin does not affect intracellular calcium (Figure 7 supplement 1A), and (4) validate consistent expression levels of GCaMP6f in different stages (Figure 3 supplement 1C, 1D). We also repeated the gliding motility assays (Figure 8D and Figure 8F) using bradyzoites purified by the same methods used for ATP analysis to match these two experiments more closely. We found the same results in previous experiments that incubation with exogenous glucose and calcium leads to the recovery of gliding motility by bradyzoites.

4. Include discussion on the role of calcium signalling in mobilization of amylopectin carbohydrate stores and activation of bradyzoites in vivo

This is an excellent suggestion and we have included it in the Discussion.

Reviewer #1 (Recommendations for the authors):It remains unclear whether repression of calcium signalling in bradyzoites directs these stages into metabolic quiescence, or conversely, that entry into metabolic quiescence decreases the cellular energy balance with concomitant decrease in calcium uptake, depletion of intracellular stores and signalling. Can the authors draw out arguments for one side or the other?

This is an interesting point, but the current data do not argue strongly for one possibility over the other. We have added this statement to the discussion to highlight this as an interesting area for further research: "One aspect that is not addressed by our studies is whether altered expression of calcium channels and pumps is responsible for reducing energy levels and hence driving quiescence, or whether the altered calcium pathways are a consequence of initial changes in energy production." As noted above, incubation of artificially released bradyzoites in calcium plus or minus a carbon source leads to increased motility and ATP levels. However, it is likely that activation of bradyzoites will occur earlier in vivo (e.g. prior to, or during the process of egress). During natural egress, bradyzoites will be more dependent on intracellular carbohydrate stores rather than exogenous sugars. Mobilization of internal stores is suggested by the finding that calcium alone is sufficient to activate motility in extracellular bradyzoites. Discussion of this possibility, including reference to previous work showing that calcium dependent kinases regulate amylopectin degradation is warranted.

This is an excellent suggestion and we have included it in the Discussion.

Figure 8F. Can the authors provide information on the basal levels of ATP/ADP/AMP in tachyzoites suspended in the same EC buffer 10min/1hr, to allow comparison with bradyzoite data?

We thank the reviewer for this suggestion. We measured the ATP/ADP/AMP levels in tachyzoites treated with EC buffer for 10 min vs.1 hr and found that treatment did not significantly affect the ATP and ADP levels in tachyzoites but led to increase in AMP levels. These results indicate that extracellular tachyzoites do not depend on uptake of exogenous glucose for ATP production during the first hr they are extracellular, a result that is consistent with the conclusion in a previous publication (Lin SS, Blume M, von Ahsen N, Gross U, Bohne W. Extracellular *Toxoplasma gondii* tachyzoites do not require carbon source uptake for ATP maintenance, gliding motility and invasion in the first hour of their extracellular life. Int J Parasitol. 2011 Jul;41(8):835-41. doi: 10.1016/j.ijpara.2011.03.005. Epub 2011 Apr 7. PMID: 21515276.)

Now we have replaced the original figures related to ATP/ADP/AMP quantification with new Figure 8G, 8H and 8I.

Reviewer #2 (Recommendations for the authors):The introduction uses language that is a bit too definitive in places. I suggest to tone down.– I would not say that ER is the known storage site of ca^2+^, I'd say the evidence suggests it is likely, but it is far from being definitive.

Revised as suggested.

– Line 346- I would not say 1hr is a rapid change, especially in context of persistence in the brain or transmission (see below).

Revised as suggested.

There are some spelling/grammatical mistakes; for example, Line 80: Accumulation of ca^2+^ in the ER cannot lead to Mn secretion – this must be a mistake.

Revised as suggested.

I think the discussion is too long and goes over a lot of what is already presented in Results section. I recommend reducing this but instead discuss more about how these resultsimpact our understanding of transmission and persistence of a latent infection. For example, how do the lower levels of ca^2+^, motility and microneme secretion impact cyst lysis and re-invasion? Are these parameters enough or is there a transition period where bradyzoites become more motile to? Likewise, for transmission, it seems unlikely that an extracellular bradyzoite could hang around in the gut for 1 hr, to bolster energy and ca^2+^ needed to actively invade cells. Does this mean that high ca^2+^ and ATP (similar levels to tachyzoites) are not needed for transmission or do the authors think there is other possibilities? These considerations of the physiological relevance are important to discuss.

We prefer to keep the summary in the Discussion as it provides a synthesis for the many different experimental results that are presented. However, we agree that it would be beneficial to expand the text to address the issue of how altered calcium signaling in bradyzoites impacts transmission and we have now included this topic in the Discussion.